Process-based diagnostics using atmospheric budget analysis and nudging technique to identify sources of model systematic errors

Chihiro Matsukawa<sup>1,2</sup>, José M. Rodríguez<sup>2</sup>, and Sean F. Milton<sup>3</sup>

<sup>1</sup>Japan Meteorological Agency, Tokyo, Japan

<sup>2</sup>Met Office, Exeter, UK

<sup>3</sup>University of Leeds, Leeds, UK

**Correspondence:** Chihiro Matsukawa (chi matsukawa@met.kishou.go.jp)

Abstract. Identifying sources of model systematic errors is a fundamental step to successfully reduce them in general circulation models by improving the representation of relevant physical processes. In this study, we examine model error sources in the Met Office Unified Model at numerical weather prediction timescale by the combined use of two diagnostics: 1) the relaxation or "nudging" in which wind and/or temperature fields are relaxed back towards analyses throughout the simulations, and 2) atmospheric zonal-mean zonal momentum and thermal budgets. The budget analysis quantifies resolved processes and subsequently estimates unresolved processes as a residual, corresponding to model dynamics and physics, respectively. This correspondence is demonstrated by a direct comparison between the budgets and the model tendencies. A systematic error addressed in this paper is the Northern Hemisphere mid-latitude zonal wind bias in the lower stratosphere in boreal winter, characterized by an initial easterly bias that subsequently develops as a westerly bias. The momentum and thermal budget analysis for control and nudging experiments indicates that a mechanical forcing predominantly from parametrized gravity wave drag causes the easterly error and an overly strong temperature gradient around the tropopause is one of the main sources of the westerly error through the Coriolis forcing. The relevant warm bias over the tropical tropopause is mainly attributed to the budget residual term that corresponds to a thermal forcing dominated by radiative processes. This is consistent with the experimental result that temperature nudging over the tropical tropopause significantly reduces the westerly wind bias.

15 Copyright statement. The works published in this journal are distributed under the Creative Commons Attribution 4.0 License. This licence does not affect the Crown copyright work, which is re-usable under the Open Government Licence (OGL). The Creative Commons Attribution 4.0 License and the OGL are interoperable and do not conflict with, reduce or limit each other.

#### 1 Introduction

A model systematic error, or bias, is a deviation of the mean model states from the corresponding mean observed states, affecting not only local regions but also other regions remotely through associated teleconnections. Continuous enhancements incorporated into general circulation models (GCMs), such as increasing horizontal and vertical resolutions, improving representations of dynamical and physical processes, and incorporating new physical components, have massively contributed to a

significant reduction of model systematic errors in numerical weather predictions (NWP) and climate projections (e.g., Phillips et al., 2004; Bauer et al., 2015). However, further research and developments to reduce model systematic errors are still essential. Identifying origins and sources of model biases is a key step to find out how to improve model process representations to reduce the errors. It is difficult to achieve this goal using long-timescale climate simulations because of interactions of locally generated and remotely forced errors, nonlinear interactions and feedbacks among variables and various physical processes, and possible compensating errors consisting of two or more substantial errors which cancel out each other. Previous studies show that many long-timescale errors develop within the first few days of simulations and the fastest growing errors are probably associated with the model physics (Martin et al., 2010; Ma et al., 2014; Martin et al., 2021). A better understanding of the initial error growth, when errors are more locally forced and models are constrained by data assimilation through initial conditions, yields insights into the relevant sources of model errors at the physical process level (Rodwell and Palmer, 2007). In addition, there has been a wide variety of diagnostic methods to evaluate model systematic errors: for instance, potential vorticity budget diagnostics (e.g., Chagnon et al., 2013; Saffin et al., 2016), semi-geostrophic balance tool (e.g., Sánchez et al., 2020), perturbed parameter ensemble technique (e.g., Sexton et al., 2019; Karmalkar et al., 2019; Williams et al., 2020), single-column models experiments (e.g., Duynkerke et al., 2004; Lenderink et al., 2004; Svensson et al., 2011), model intercomparison projects (e.g., Elvidge et al., 2019; van Niekerk et al., 2020), and WGNE conferences on model systematic errors (e.g., Frassoni et al., 2023).

The relaxation or "nudging" method is a widely used practical framework that involves the addition of artificial terms to the prognostic equations to relax some of the model variables towards given (usually observed or analysed) states throughout the integration (Jeuken et al., 1996). Nudging forcing towards analysed states is comparable to the forecast error at a particular timescale with the reversed signs. Hence, it is beneficial to evaluate the nudging forcing itself to explore the sources of the model systematic errors. In addition, a nudging technique applied to selected regions or model levels, namely "regional nudging", has also been used for the purpose of understanding their remote influences on other domains and diagnosing origins of forecast errors (e.g., Klinker, 1990; Hoskins et al., 2012; Rodríguez and Milton, 2019).

45

Another well-established technique is to diagnose atmospheric momentum and thermal budgets in analyses and forecasts. The budget equations describe a time evolution of wind and temperature field and individual contributions from resolved and unresolved processes. In particular, unresolved processes including a mechanical forcing in the momentum budget and a diabatic heating in the thermal budget can alter the time evolution of the corresponding variable as a source or sink term of the prognostic equations. Better representations of these processes, which need to be parametrized in GCMs, are essential for obtaining more accurate predictions (Bauer et al., 2015). An indirect method of quantifying unresolved forcing is to estimate forcing as a residual of the budget equation using observed or analysed data (e.g., Yanai et al., 1973; Hartmann, 1976; Hamilton, 1983; Smith and Lyjak, 1985; Palmer et al., 1986; Holopainen, 1987). This method has been utilized for model diagnostics in some previous studies to identify possible deficiencies in physics parametrizations. Klinker and Sardeshmukh (1992) defined the balance requirement as the sum of all the adiabatic terms in the zonal-mean zonal flow tendency equation with the sign reversed. The comparison of the balance requirement deduced from the analysis data with parametrized tendencies can suggest possible deficiencies in the model physics to balance the sum of the adiabatic terms. Milton and Wilson (1996) applied the same diagnostic as Klinker and Sardeshmukh (1992) and indicated systematic deficiencies in parametrized tendencies. They

demonstrated that the incorporation of the new parametrizations associated with subgrid-scale orography leads to better initial momentum balances as well as reduced systematic errors in the general circulation. As an analogous diagnostic, van Niekerk et al. (2016) used an angular momentum budget analysis to examine a sensitivity of the resolved and parametrized surface drag to changes in horizontal resolution and parametrization. Their approach was to use the nudging framework to constrain the contribution from the angular momentum flux convergence (AMFC), and to determine the contributions from the resolved mountain torque and parametrized surface torque by balancing the AMFC. They found that a parametrized orographic torque in their model was excessive at lower resolutions.

65

The present study is aimed at understanding mechanisms of forecast error growth in global Met Office Unified Model (hereafter MetUM) at NWP timescales, and finding possible sources of the model errors. We focus on the Northern Hemisphere (NH) mid-latitude zonal wind errors from the upper troposphere to the stratosphere in boreal winter. Many studies have been dedicated to understanding mechanisms of the model systematic errors in the different version of the MetUM, particularly for a temperature bias around the tropopause (Hardiman et al., 2015; Bland et al., 2021). Hardiman et al. (2015) have executed several sensitivity experiments to examine the tropical tropopause warm bias that indicate that microphysical and radiative processes influence the temperature. Bland et al. (2021) have investigated a connection between a cold bias and a moist bias in the extratropical lowermost stratosphere through long-wave radiative effects. The approach adopted in this study is to analyse the atmospheric zonal-mean zonal momentum and thermal budgets in initialized NWP hindcasts and various nudging experiments. We use the globally nudged simulation to provide a best possible estimate of the truth to validate individual components of the momentum and thermal budgets in the MetUM. The budgets are examined to identify which component has a dominant contribution to initial and subsequent error growth through their comparison between the non-nudged control experiment and the globally nudged experiment. In addition, we apply the budget analysis to the regional nudging experiments to examine the impact of regional forcing on the general circulation patterns as well as the specific model errors. A combined application of the momentum and thermal budget analysis enables us to obtain information on wind circulation and temperature interactions and possible error compensations.

This paper is organized as follows. Section 2 describes the model and nudging framework used in this study, experiments carried out, and the details of the budget analysis. Section 3 presents a general view of the model systematic errors and highlights the NH mid-latitude zonal wind errors in boreal winter. Section 4 shows zonal-mean zonal momentum and thermal budgets in control and global nudging experiments and demonstrates a correspondence between budget components and model tendencies. In Sect. 5, we address the NH mid-latitude lower-stratospheric zonal wind errors and investigate sources of the errors using the budget analysis. Section 5 also examines remote and indirect impacts of regional nudging and its momentum budget. Finally, Sect. 6 summarizes the main conclusions and discusses further potential applications of the diagnostics and possible model error sources.

# 2 Methodology

In this section, we describe details of the global MetUM used in this study, the nudging framework in the MetUM, the experimental design, and the zonal-mean zonal momentum and thermal budget analysis.

### 2.1 Model description

The MetUM is a numerical model which has been developed for use in regional and global simulations across weather to climate timescales (Cullen, 1993; Senior et al., 2011; Brown et al., 2012). Its scientific configuration used in this study is a global atmosphere and land configuration with a version of Global Atmosphere Land 9 (hereafter GAL9, in prep.; containing incremental upgrades from the former versions (e.g., Walters et al., 2019)), which is uncoupled with ocean and sea-ice dynamical model. The dynamical core, ENDGame, adopts a semi-implicit semi-Lagrangian formulation to solve the non-hydrostatic, fully compressible deep-atmosphere equations of motion (Wood et al., 2014). The atmospheric prognostic variables are zonal, meridional, and vertical wind velocity, dry virtual potential temperature, Exner pressure, dry density, and moist prognostic variables such as mixing ratio of moisture variables (water vapour, cloud water, and cloud ice) and cloud prognostic fields. Physical processes which are not represented or not resolved in the dynamical core, such as friction, condensation and evaporation, radiative heating and cooling, and too small-scale phenomena to be resolved at the grid scale, are accounted for by parametrizations. Parametrizations employed by the MetUM include shortwave and longwave radiation (Edwards and Slingo, 1996), microphysics (Wilson and Ballard, 1999), gravity wave drag consisting of non-orographic gravity wave (Scaife et al., 2002) and sub-grid scale orographic drag (Appendix in Vosper (2015) for details), convection (Gregory and Rowntree, 1990), turbulent mixing represented by the boundary-layer scheme (Lock et al., 2000), and large-scale cloud (Wilson et al., 2008a, b). Processes at the land surface and in the subsurface soil are represented by a community land surface model, the Joint UK Land Environment Simulator (JULES; Best et al., 2011). Contributions of the model dynamics and the individual physics parametrizations to the time evolution of prognostic variables can be diagnosed using increments per model time step, or tendencies that are equivalent to the increments per unit time.

Different applications of the MetUM across a wide range of temporal and spatial scales employ essentially the same model configuration, such as dynamical core and physics parametrizations. The MetUM used in this study has a horizontal resolution of N320 grid (0.5625°longitude × 0.375°latitude; approximately 40 km in the midlatitudes) with 70 vertical levels extending to 80 km altitude. The forecast model time step is 12 minutes. The horizontal resolution is lower than that of the Met Office's operational global deterministic NWP model (N1280 grid; approximately 10 km horizontal resolution in the midlatitudes). Under the across-scale approach, we adopt a moderate horizontal resolution to investigate large-scale model systematic errors, which provides benefits with regard to computational resources. However, since representations of the dynamical core and the physics parametrizations may have a sensitivity to model horizontal resolutions, extending our analysis to different resolutions of the MetUM may also be required.

### 120 **2.2** Nudging

The nudging technique with Newtonian relaxation is a method that relaxes predicted variables of GCMs back towards given meteorological fields by adding an unphysical relaxation term to the prognostic equations (Jeuken et al., 1996). The nudging process incorporated into the MetUM at the very end of each model time step is written as follows (Telford et al., 2008; van Niekerk et al., 2016):

125 
$$X_F = X_M + \frac{\delta t}{\tau} (X_A - X_M)$$
 (1)

where X is the prognostic model variable at the current model time step,  $\delta t$  is the time interval of the model integration (i.e., 12 minutes in this study), and  $\tau$  is the relaxation timescale of nudging. Subscripts F, M, and A denote the variables after nudging, those after dynamics and physics calculations just before nudging, and those used as a nudging forcing, respectively. In this study, the 6-hourly operational MetUM analysis at N1280 grid is regridded onto the model resolution (i.e., N320 grid) and then used as a forcing. Since the analysis data are available every 6 hours, they are linearly interpolated into each model time step. The choice of the relaxation timescale is arbitrary. The shorter the relaxation timescale is, the more strongly the model prognostic variables are relaxed towards forcing data (i.e. MetUM analysis data). Too long relaxation timescale is ineffective, and too short relaxation timescale could make the model unstable (Telford et al., 2008). We select relaxation timescale used in this study to be 6 hours which corresponds to the temporal spacing of the MetUM analysis. Notice that nudging is applied throughout model integration regardless of the relaxation timescale.

Additional increments due to nudging, which are expressed by the second term of the right-hand side of Eq. (1), are calculated as a difference of the variables between forcing data  $X_A$  and the constrained model predictions  $X_M$  within the nudged run multiplied by the relaxation coefficient. Therefore, the nudging increments are comparable to the forecast errors at least those growing over the timescale of the relaxation. Since the nudging forcing alters the time evolution of nudged prognostic variables alongside the model dynamics and physics, the nudging tendencies are equivalent to the increments per unit time and are comparable with the model tendencies due to the dynamics and physics.

The prognostic model variables X which can be constrained by nudging are zonal wind, meridional wind, and potential temperature. Other prognostic variables, such as vertical wind velocity and mixing ratio of moisture variables, are allowed to evolve freely and respond to the forced variables. Nudged domains and model levels can be prescribed arbitrarily and are accompanied by interfacing transition zones and layers to ensure a smooth transition between the nudged and free-running parts of the simulation. The transition zones and layers are smoothed using the hyperbolic tangent function over 10 degrees in the horizontal and the linear function over two model levels in the vertical.

# 2.3 Experimental design

To examine model error growth in boreal winter at NWP timescale, deterministic 15-day forecasts are started at 00:00 UTC between 16 November 2018 and 27 February 2019 and evaluated over the period from December 2018 to February 2019 (90 cases in total). A series of the operational global MetUM analyses produced by the data assimilation system, Hybrid-4DVar

**Table 1.** Control and nudging experiments executed. U, V, and  $\Theta$  in nudged variables indicate zonal wind velocity, meridional wind velocity, and potential temperature, respectively.

| Nudged domains                                                      | Nudged model levels                                                                                                                                         | Nudged variables                                                                                                                                                                                                                                                                                                                                                                                                                                                                                          |
|---------------------------------------------------------------------|-------------------------------------------------------------------------------------------------------------------------------------------------------------|-----------------------------------------------------------------------------------------------------------------------------------------------------------------------------------------------------------------------------------------------------------------------------------------------------------------------------------------------------------------------------------------------------------------------------------------------------------------------------------------------------------|
| N/A                                                                 | N/A                                                                                                                                                         | N/A                                                                                                                                                                                                                                                                                                                                                                                                                                                                                                       |
| Globe $(0^{\circ}-360^{\circ}, 90^{\circ}S-90^{\circ}N)$            | Whole model levels                                                                                                                                          | $U, V, \Theta$                                                                                                                                                                                                                                                                                                                                                                                                                                                                                            |
| Globe $(0^{\circ}-360^{\circ}, 90^{\circ}S-90^{\circ}N)$            | Whole model levels                                                                                                                                          | $\Theta$                                                                                                                                                                                                                                                                                                                                                                                                                                                                                                  |
| NH Tropics $(0^{\circ}-360^{\circ}, 0^{\circ}-30^{\circ}N)$         | Whole model levels                                                                                                                                          | $\Theta$                                                                                                                                                                                                                                                                                                                                                                                                                                                                                                  |
| NH Tropics $(0^{\circ}-360^{\circ}, 0^{\circ}-30^{\circ}N)$         | Around the tropopause (approx. 200-70 hPa)                                                                                                                  | $\Theta$                                                                                                                                                                                                                                                                                                                                                                                                                                                                                                  |
| NH High Latitude $(0^{\circ}-360^{\circ}, 60^{\circ}N-90^{\circ}N)$ | Whole model levels                                                                                                                                          | $\Theta$                                                                                                                                                                                                                                                                                                                                                                                                                                                                                                  |
| NH High Latitude ( $0^{\circ}$ –360°, $60^{\circ}$ N–90°N)          | Around the tropopause (approx. 300-100 hPa)                                                                                                                 | Θ                                                                                                                                                                                                                                                                                                                                                                                                                                                                                                         |
|                                                                     | N/A Globe (0°-360°, 90°S-90°N) Globe (0°-360°, 90°S-90°N) NH Tropics (0°-360°, 0°-30°N) NH Tropics (0°-360°, 0°-30°N) NH High Latitude (0°-360°, 60°N-90°N) | N/A N/A  Globe $(0^{\circ}-360^{\circ}, 90^{\circ}\text{S}-90^{\circ}\text{N})$ Whole model levels  Globe $(0^{\circ}-360^{\circ}, 90^{\circ}\text{S}-90^{\circ}\text{N})$ Whole model levels  NH Tropics $(0^{\circ}-360^{\circ}, 0^{\circ}-30^{\circ}\text{N})$ Whole model levels  NH Tropics $(0^{\circ}-360^{\circ}, 0^{\circ}-30^{\circ}\text{N})$ Around the tropopause (approx. 200–70 hPa)  NH High Latitude $(0^{\circ}-360^{\circ}, 60^{\circ}\text{N}-90^{\circ}\text{N})$ Whole model levels |

(Clayton et al., 2013) based on GA6.1 configuration (Walters et al., 2017) operational in 2018/2019, is spatially interpolated to the model resolution and used as the initial conditions. Lower boundary conditions are given by the sea surface temperature and sea ice concentration of the Operational Sea Surface Temperature and Ice Analysis (OSTIA; Donlon et al., 2012) products fixed at the field on initial dates throughout the 15-day simulations.

The momentum and thermal budgets described below in Sect. 2.4 and model tendencies due to dynamics and individual parametrizations (and nudging forcing) are calculated using the experimental data. The budgets are analysed at the same horizontal resolution as the model on 28 pressure levels (i.e., 1000, 950, 925, 850, 700, 600, 500, 400, 300, 250, 200, 150, 100, 70, 50, 30, 20, 10, 7, 5, 3, 2, 1, 0.7, 0.5, 0.3, 0.2, 0.1 hPa) with masking of the levels below the ground. On the other hand, model tendencies are evaluated at the 70 model native levels. Temporal-mean variables required in the analysis of the budgets and the model tendencies are calculated from the fields at each time step in the model.

Control and various nudging experiments executed in this study are summarized in Table 1. We apply momentum and thermal budget analysis (see details in Sect. 2.4) to the control experiment referred to as CNTL and the global nudging experiment referred to as GLN. We consider individual budget components of GLN as the best possible estimate of truth instead of analysis data which is unable to provide the temporal-mean variables required in this study. This is why the reasonably short relaxation timescale is selected in the nudging experiments. Nudging frameworks which constrain a part of the variables could bring a better understanding of feedbacks among variables and disentangle compensating errors specifically between horizontal wind and temperature (Wehrli et al., 2018). In addition, regional nudging, in which the model variables are relaxed back towards the analyses over subdomains where there might be significant systematic errors, potentially provide experimental insights into origins of the errors. We performed another global nudging experiment in which only temperature is constrained over the globe, referred to as GLNT, and nudging sensitivity experiments in which temperature is constrained over different domains at various altitudes (i.e., NHTrpT, NHTrpTrpT, NHHLT, and NHHLTrpT; see Table 1 in detail) to test how much wind biases in CNTL could be influenced by temperature biases and diagnose the remote impact of nudging. Note that the tropopause

temperature nudging in NHTrpTrpT and NHHLTrpT is applied in a different range of pressure levels depending on their latitudinal positions.

## 2.4 Atmospheric zonal momentum and thermal budgets

The framework of the atmospheric zonal-mean momentum and thermal budgets is a well-established diagnostic tool for examining the contribution of resolved and unresolved processes to the large-scale structure of the atmosphere. Based on the primitive equations, the zonal-mean zonal momentum and thermal budget equations in spherical and pressure coordinates are derived as below (Hartmann, 1976; Andrews, 1987):

$$\frac{\partial [u]}{\partial t} = -\left(\frac{[\overline{v}]}{a\cos\phi} \frac{\partial [\overline{u}]\cos\phi}{\partial \phi} + [\overline{\omega}] \frac{\partial [\overline{u}]}{\partial p}\right)$$
Mean Stationary Flow Advection
$$-\left(\frac{1}{a\cos^2\phi} \frac{\partial [\overline{u}^*\overline{v}^*]\cos^2\phi}{\partial \phi} + \frac{\partial [\overline{u}^*\overline{w}^*]}{\partial p}\right)$$
Stationary Eddy Component
$$-\left(\frac{1}{a\cos^2\phi} \frac{\partial [\overline{u}'\overline{v}^*]\cos^2\phi}{\partial \phi} + \frac{\partial [\overline{u}'\overline{w}^*]}{\partial p}\right)$$
Transient Eddy Component
$$+f[\overline{v}] + [F_u]$$
Coriolis Residual
$$\frac{\partial [\overline{T}]}{\partial t} = -\left(\frac{[\overline{v}]}{a} \frac{\partial [\overline{T}]}{\partial \phi} + [\overline{\omega}] \frac{\partial [\overline{T}]}{\partial p} - \frac{R_d}{pc_p} [\overline{\omega}] [\overline{T}]\right)$$
Mean Stationary Flow Component
$$-\left(\frac{1}{a\cos\phi} \frac{\partial [\overline{v}^*\overline{T}^*]\cos\phi}{\partial \phi} + \frac{\partial [\overline{\omega}^*\overline{T}^*]}{\partial p} - \frac{R_d}{pc_p} [\overline{w}^*\overline{T}^*]\right)$$
Stationary Eddy Component
$$-\left(\frac{1}{a\cos\phi} \frac{\partial [\overline{v}^*\overline{T}^*]\cos\phi}{\partial \phi} + \frac{\partial [\overline{\omega}^*\overline{T}^*]}{\partial p} - \frac{R_d}{pc_p} [\overline{w}^*\overline{T}^*]\right)$$
Transient Eddy Component
$$+\frac{[\overline{Q}]}{c_p}$$
(3)

where u and v are zonal and meridional components of wind velocity,  $\omega$  is vertical pressure velocity, a is the mean radius of Earth,  $\phi$  is latitude, p is pressure, f is the Coriolis parameter, T is temperature,  $R_d$  is the gas constant,  $c_p$  is the specific heat of air at constant pressure,  $F_u$  is a mechanical forcing term of the zonal momentum equation, and Q is a diabatic heating rate. Overbars and primes denote the temporal mean and departure from the temporal mean, respectively, and square brackets and

asterisks denote the zonal mean and departure from the zonal mean, respectively. The temporally and zonally averaged flow (e.g.,  $[\overline{u}]$ ) is referred to as mean stationary flow, and the departure from the temporal mean (e.g.,  $u' = u - \overline{u}$ ) is referred to as transient eddy, and the departure of the temporally averaged variable from its zonal mean (e.g.,  $\overline{u}^* = \overline{u} - [\overline{u}]$ ) is referred to as stationary eddy.

The right-hand side of Eq. (2), in order, represents an advection of mean zonal wind by mean stationary flows, a convergence of stationary eddy momentum fluxes, that of transient eddy momentum fluxes, Coriolis forcing, and a residual term. The transient and stationary eddy fluxes are evaluated, for instance, as  $\overline{u'v'} = \overline{uv} - \overline{u}\overline{v}$  and  $[\overline{u}^*\overline{v}^*] = [\overline{u}\overline{v}] - [\overline{u}][\overline{v}]$ , respectively. In Eq. (3), the right-hand side, from left to right, shows a warming/cooling due to advection of mean temperature by mean stationary flow and adiabatic heating with mean vertical motions, a convergence and an energy conversion of stationary eddy heat flux, those of transient eddy heat flux, and a residual term. The terms other than the residual term are interpreted as contributions from resolved dynamical components and are evaluated directly from the forecast data specifically  $u, v, \omega$ , and T fields. As in the manner in Martineau et al. (2016), a second-order centered difference scheme is used for calculating meridional and vertical derivatives, while meridional derivatives over the poles and vertical derivatives at the top or bottom of the pressure levels require a first-order difference with their adjacent grids.

The last term of right-hand side of Eq. (2, 3), determined as a residual of the other terms including the left-hand side and referred to as residual term, represents contributions of the processes which are not represented by the resolved processes. In the partial differential equations of Eq. (2, 3), the residual term corresponds to a source or sink of the momentum and thermodynamic equations which consists of the frictional forcing and the diabatic heating due to radiative forcing, latent heat releases, and latent and sensible heat fluxes at the surface. On the other hand, the residual term diagnosed using spatially discretized data could include effects of subgrid processes, which are too-small scale to be resolved in a grid scale, on a grid-box mean field as well as the frictional or diabatic forcing. Ideally, the residual term should be qualitatively equal to the total parametrized forcing calculated in the model (Holopainen, 1987), although this is not necessarily the case as described in Sect. 2.5. It is noteworthy that "residual" in this study, which is equivalent to estimated unresolved tendencies, is defined differently from "residual" termed in some previous studies (e.g., Klinker and Sardeshmukh, 1992; Milton and Wilson, 1996), which is the difference between estimated unresolved tendencies in the analysis and parametrized physics tendencies.

There are several options of how to define and compute temporal-mean fields denoted by overbars in Eq. (2, 3), and it changes what the temporal derivative term on the left-hand side means. In this study, overbars represent the temporal averaging over the given forecast length in each case, not over the 3 months of the experimental period. Thus, a temporal-mean field over the forecast length (e.g.,  $\overline{u}$ ) varies among experiments and also among cases. The temporal average over the forecast length is computed using the fields at each model time step, which is the reason why globally nudged simulation data instead of analysis data is used to calculate individual budget components for validations because the analysis data has only 6-hourly instantaneous fields. This definition of temporal mean allows the budget equation to describe a time evolution over a particular forecast length in a single case, which makes it straightforward to diagnose forecast error growth at NWP timescale. The left-hand side of the Eq. (2, 3) indicating a total tendency of the variable is evaluated as a temporal change from initial conditions to model states at a given forecast lead time.

The momentum and thermal budget analysis is applied to each simulated case individually, and then averaged over the cases if mean budgets are examined. In this context, the terms "mean flow" and "transient eddy" in this paper should be interpreted cautiously because the methodology is different from that commonly used in climatological studies (e.g., Peixoto and Oort, 1992). The partition of the nonlinear advective term among three components including mean stationary flow, stationary eddies, and transient eddies becomes slightly more complex and critically depends on the forecast lead time in this study. For example, in the momentum budget from an initial condition to a forecast at Day 1, if a migratory cyclone or anticyclone is simulated to be quasi-stationary throughout that time period, it is identified as the stationary eddy rather than the transient eddy. Therefore, the contribution of the transient eddy flux becomes smaller at a shorter forecast lead time, and the mean stationary flow and the stationary eddy components are responsible for the remaining part. Although the Coriolis forcing term involves the mean stationary flow and constitutes a part of the resolved processes, it can be independently evaluated regardless of the forecast length because of its linearity. The partition should depend not only on the definition of the temporal mean but also on the horizontal and vertical resolution of the model (and data) and the configuration of the model dynamics and physics.

## 2.5 Atmospheric budgets and model tendencies

Whilst the model dynamics and physics tendencies themselves, calculated in the model online, are not used for the offline budget calculation in our framework, the diagnosed resolved processes and the residual term correspond to the model dynamics and physics tendencies, respectively. A comparison of two diagnostics, atmospheric budgets and model tendencies, is helpful in understanding their physical meaning. The budget analysis diagnosed using forecast data on pressure levels quantifies contributions to the total tendencies from the resolved processes and the unresolved processes. On the other hand, model tendencies of prognostic variables are calculated by dynamical core and physics parametrizations in numerical models at each model time step during their time integration at each model level. Individual components of both atmospheric budgets and model tendencies represent contributions of the processes in the respective framework to the total tendency of the corresponding variable. Therefore, the atmospheric budget diagnostics can be compared with the model tendencies to demonstrate a qualitative similarity between the unresolved term in the budget equation and the model tendencies of the physics parametrizations and between the resolved term and the tendencies of the model dynamics.

The correspondence between the budget diagnostics and the model tendencies is expected from the derivation of the budget equation, but they are not necessarily exactly equivalent. The residual term could contain effects from imbalances caused by nudging increments (in the case of nudging experiments) and computational errors through the interpolation of forecast fields from model levels to pressure levels and the numerical methods employed to evaluate the budgets (Martineau et al., 2016) as well as the unresolved processes. A discrepancy between the primitive equations used in the budget diagnostics and the governing equations of the MetUM could also generate a difference between the resolved processes and the model dynamics tendencies and therefore between the residual term and the parametrization tendencies to some extent. In the case of nudging experiments, forcing due to nudging, which is associated with the very short timescale model error in the MetUM, should also constitute a part of the residual term.

# 2.6 Validation of atmospheric budgets

260

265

To validate the budget components in CNTL and identify which components contribute to a particular forecast error growth, we use the nudging technique to create quasi-analysis data and calculate the best possible estimate of truth of atmospheric budgets. An error of a given forecast variable against analysis in a single case, defined as its difference between forecast and analysis, is proportional to an error of total tendency as follows:

$$x_{e}(t_{i}; \Delta t) \equiv x_{f}(t_{i}; \Delta t) - x_{a}(t_{i} + \Delta t)$$

$$= \left\{ \overline{\left(\frac{\partial x}{\partial t}\right)_{f, t_{i}}} - \overline{\left(\frac{\partial x}{\partial t}\right)_{a, t_{i}}} \right\} \Delta t$$
(4)

where x is a given variable,  $t_i$  is an initialized time in a given single case, and  $\Delta t$  is a forecast length. The subscripts e, f, and a denote error, forecast, and analysis, respectively. The overbars are the same as defined in Eq. (2, 3). In this study, the total tendency of the analysis averaged over a given forecast length is approximated by that of GLN, and we refer to a difference in budget components between CNTL and GLN as "quasi-error". As a result, the validation of the momentum and thermal budgets can decompose a quasi-error in the total tendencies into individual budget components.

The atmospheric budgets of GLN should be interpreted with caution because they have an error against the analysis. Firstly, nudged variables in nudging experiments still have errors against the MetUM analysis. The nudging in some cases is not able to relax sufficiently towards the forcing data with a moderate relaxation timescale especially if the error grows rapidly at a very short timescale. In addition, the vertical velocity, required by the budget analysis but not directly nudged, is driven by the horizontal wind and temperature nudging forcing. Therefore, there should be a non-negligible error in the individual budget components in GLN, making it difficult to precisely evaluate the residual term as a consequence. Secondly, since the 6-hourly MetUM analyses used as a forcing data of the nudging are linearly interpolated into each model time step as part of the nudged simulation methodology, the temporal interpolation leads to an underestimation of the intra-6-hour variability, and therefore results in underestimated transient eddy components and consequently overestimated residual terms. Although these deficiencies might affect a quasi-error in budget components of CNTL against GLN to some extent, it is reasonable to use GLN as the best estimate of truth.

The interpretation of differences in the residual term among experiments is slightly complex. It is attributed, by definition, to differences in the other resolved terms because the residual term in each experiment is determined by the sum of the other terms to close the budget. From another perspective, the residual term is expected to correspond to the sum of the physics parametrization tendencies and the nudging forcing. The quasi-error in the residual term against GLN could provide some implications for deficiencies in the parametrizations through a comparison between the residual term and the individual parametrization tendencies.

Unlike the budget diagnostics, a difference in individual model tendencies between CNTL and GLN does not show a quasierror in the model tendencies. The physics parametrization schemes themselves used in CNTL and GLN are identical and have substantial deficiencies in principle. The parametrization tendencies in GLN are calculated using the deficient scheme with nudged grid-box mean fields which are much closer to the analysis than CNTL. The difference in parametrization tendencies

**Figure 1.** Zonal-mean errors of the CNTL forecast at a forecast lead time of (a, b) Day 1, (c, d) Day 5, and (e, f) Day 15 averaged over the 90 cases from 1 Dec. 2018 to 28 Feb. 2019 in (a, c, e) zonal wind velocity [m s<sup>-1</sup>], and (b, d, f) temperature [K]. Contour indicates the forecasts, and colour indicates mean errors against the MetUM analysis. The black horizontal lines A, B, and C are inserted for later reference (Figs. 2, 12, 13, and 14; see the text in detail).

between two experiments represents a difference in the responses of the parametrization scheme to different grid-box mean fields.

**Figure 2.** (a–c) Timeseries of the MetUM analysis and the CNTL and GLN forecasts of the individual cases up to Day 7 for (a) area-averaged zonal wind velocity [m s $^{-1}$ ] over 30°N-60°N at 50 hPa, (b) area-averaged temperature [K] over 0°-30°N at 70 hPa, and (c) area-averaged temperature [K] over 60°N-90°N at 200 hPa. These features are depicted by the line A, B, and C in Fig. 1, respectively. (d) Time evolution of NH mid-latitude (30°N-60°N) 50 hPa zonal wind errors [m s $^{-1}$ ] against the MetUM analysis, shown in (a), averaged for DJF 2018-2019. Error bars indicate a 95 % confidence interval estimated using the bootstrap method with 10,000 resamples.

### 3 Model systematic errors

Prior to analysing the momentum and thermal budgets, we present a general view of the model systematic errors in zonal wind and temperature. Figure 1 exhibits mean errors in zonal-mean zonal wind and temperature in CNTL against the MetUM analysis in December-January-February (DJF) 2018/2019 at specific forecast lead times. The CNTL simulations up to Day 15 show a westerly wind bias in the NH and Southern Hemisphere (SH) midlatitudes from the upper troposphere to the stratosphere, a westerly wind bias in the tropical upper troposphere, a warm temperature bias around the tropical tropopause, a cold temperature bias around the tropical tropopause over the poles, and a cold temperature bias in the stratosphere. Some of the biases exist even at Day 1 and evolve steadily in the same location, such as the westerly bias in the tropical upper troposphere, the warm bias around the tropical tropopause, and the cold bias in the stratosphere. The NH mid-latitude westerly wind bias in the stratosphere appears to shift poleward gradually as a forecast lead time gets longer. Most of the pronounced errors in Fig. 1 are also discernible in 20-years low-resolution climate simulations with a minor sensitivity to horizontal resolution (not shown). These errors develop quickly in the NWP timescale and remain in the climate timescale.

The westerly wind bias in the NH mid-latitude stratosphere is consistent with the warm temperature bias in the tropical tropopause and the cold temperature bias over the poles through thermal wind relation. Although causality is unclear, an

increase in the zonal wind error with altitude balances with an excessive meridional temperature gradient arising from the temperature biases. To investigate these biases, we focus on three features highlighting the pronounced zonal-mean systematic errors: A) zonal wind in the NH mid-latitude stratosphere averaged from 30°N to 60°N at 50 hPa, B) temperature around the NH tropical to subtropical tropopause averaged from the equator to 30°N at 70 hPa, and C) temperature around the NH highlatitude tropopause averaged from 60°N to 90°N at 200 hPa. These features are illustrated by the black horizontal lines A, B, and C in Fig. 1(e, f). Figure 2 presents the zonal wind and temperature forecast error growth over these domains in the individual cases up to Day 7 (Fig. 2(a-c)) and time evolution of the zonal wind error averaged over all the cases (Fig. 2(d)). The forecasts at Day 0 obviously have no errors against the analysis because the forecasts are started from the corresponding MetUM analysis. Timeseries of the CNTL forecasts reveal the noticeable drifts of the wind and temperature in most of the cases. Zonal wind tends to increase relative to the analysis in the NH mid-latitude lower stratosphere (Fig. 2(a)). Meanwhile, temperature increases around the tropical tropopause (Fig. 2(b)) and decreases near the NH polar tropopause (Fig. 2(c)) relative to the analysis. In particular, these temperature errors develop quickly and systematically in almost all the cases and are statistically significant at the 95 % confidence level throughout the forecast lead time up to Day 15. The rapid error growth and consistent drifts suggest that these errors are relevant to the model physics rather than changes in the atmospheric circulation (Martin et al., 2010; Ma et al., 2014; Martin et al., 2021). By contrast, the NH mid-latitude zonal wind shows an easterly bias at Day 1, and it is replaced by a westerly wind bias at Day 3 and thereafter (Fig. 2(d)). This characteristic time evolution of the zonal wind error implies that the sources of the initial easterly bias and the subsequent westerly bias are different.

As expected, timeseries of the GLN forecasts are apparently almost identical with those of the MetUM analysis. This illustrates that the model variables are properly relaxed towards the forcing data, the MetUM analysis. However, there are systematic errors with a small magnitude especially in the NH mid-latitude stratospheric zonal wind (Fig. 2(a, d)). The small initial easterly wind bias in GLN continues uniformly throughout the forecasts up to Day 15, even though the zonal wind itself is nudged. Zonal wind errors in GLN vary among cases but do not depend on the forecast lead time if their valid time is the same (see below in Fig. 12(a, b)). Although this consistent error in GLN seemingly looks strange, this is reasonable because the forcing data, the MetUM analysis, which model prognostic variables relaxed towards, does not depend on forecast lead time. As a result, GLN shows the same forecast fields and therefore the same error fields against analysis at specific valid time regardless of the forecast lead time.

Spatial distributions of the model biases could also reveal error characteristics. Global spatial maps of the zonal wind bias at 50 hPa in CNTL and GLN at Day 1 and Day 5 are shown in Fig. 3. The easterly wind bias over the Himalayas, the Tibetan Plateau, and its downstream region are the main contributors to the zonally averaged wind error in the lower stratosphere (Fig. 3(a)). It becomes less obvious relative to other regions as forecast lead time gets longer (Fig. 3(c)). Whilst the nudging of zonal wind as well as meridional wind and temperature in GLN makes the error against the analysis much smaller in the spatial-distribution perspective, the easterly wind bias remains in GLN especially over the Himalayas and the Tibetan Plateau (Fig. 3(b, d)). This local easterly wind bias is probably associated with the modelled orography and/or the orographic gravity wave drag parametrization.

**Figure 3.** Spatial distributions of zonal wind errors [m s<sup>-1</sup>] at 50 hPa at a forecast lead time of (a, b) Day 1 and (c, d) Day 5 averaged over the 90 cases from 1 Dec. 2018 to 28 Feb. 2019 in (a, c) CNTL, and (b, d) GLN. Contour indicates the forecasts, and colour indicates mean errors against the MetUM analysis.

### 340 4 Zonal-mean zonal momentum and thermal budget

The atmospheric zonal momentum and thermal budget analysis specifies to what extent the individual components in the budget equation contribute to the total tendency of the corresponding variable. In this section, we establish a correspondence between these budget components and the model tendencies due to dynamics and physics. To identify which component has a dominant contribution to the forecast error growth, a quasi-error in budget components of CNTL against GLN is evaluated.

#### 345 4.1 Zonal momentum budgets

350

The atmospheric zonal-mean zonal momentum equation indicates that the total tendency of zonal-mean zonal wind is separated into contributions from the mean stationary flow advection, the stationary and transient eddy momentum flux convergence, the Coriolis forcing, and the residual term. A zonal-mean zonal momentum budget at Day 0–5 is selected here to examine the evolving errors. Figure 4 shows vertical distributions of the mean total tendency from initial conditions to T+120h and contributions from the resolved processes and the residual term averaged over the 90 cases, and their quasi-error of CNTL against GLN. The total forecast tendency in CNTL averaged over all the cases shows a considerably similar pattern to the mean error (Fig. 1(c)). This is because the analysis tendency averaged over a long experimental period becomes vanishingly small compared with the forecast tendency so that the error is approximately proportional to the total forecast tendency (Rodwell and

**Figure 4.** Latitude-height cross sections of zonal-mean (a, d) total tendency of zonal wind, and budget contributions from (b, e) the resolved processes (i.e., sum of the mean stationary flow advection term, the stationary eddy momentum flux term, the transient eddy momentum flux term, and the Coriolis term shown in Fig. 5) and (c, f) the residual term of the zonal-mean zonal momentum budget equation  $[m \ s^{-1} \ day^{-1}]$  written in Eq. (2) from initial conditions to T+120h averaged over the 90 cases (a-c) in CNTL, and (d-f) their differences between CNTL and GLN. Contour in (a), for reference, indicates the zonal wind forecast at T+120h which is identical with that in Fig. 1(c). Note that the colour scales in (a-c) and (d-f) are different.

Palmer, 2007). Therefore, the difference in the total tendency between CNTL and GLN, shown in Fig. 4(d), is almost the same as the total tendency in CNTL, shown in Fig. 4(a), whilst the color scales are different. Figure 5 shows the individual terms that constitute the resolved processes shown in Fig. 4(b, e). The mean stationary flow advection in the troposphere contributes to a decrease in zonal-mean zonal wind in the tropics with the southerly wind (i.e.,  $-\frac{[\overline{v}]}{a\cos\phi}\frac{\partial [\overline{u}]\cos\phi}{\partial \phi}<0$  because  $[\overline{v}]>0$  and  $\frac{\partial [\overline{u}]\cos\phi}{\partial \phi}>0$ ), a decrease in zonal-mean zonal wind in the mid-latitude with the northerly wind (i.e.,  $-\frac{[\overline{v}]}{a\cos\phi}\frac{\partial [\overline{u}]\cos\phi}{\partial \phi}<0$  because  $[\overline{v}]<0$  and  $\frac{\partial [\overline{u}]\cos\phi}{\partial \phi}<0$ ), and an increase in zonal-mean zonal wind in the subtropics with the descents (i.e.,  $-[\overline{w}]\frac{\partial [\overline{u}]}{\partial p}>0$  because  $[\overline{w}]>0$  and  $\frac{\partial [\overline{w}]}{\partial p}

**Figure 5.** Latitude-height cross sections of a breakdown of the resolved processes (shown in Fig. 4(b, e)) to the individual terms of the zonal-mean zonal momentum budget equation  $[m \ s^{-1} \ day^{-1}]$  written in Eq. (2) (i.e., from left to right, (a, e) the mean stationary flow advection term, (b, f) the stationary eddy momentum flux term, (c, g) the transient eddy momentum flux term, and (d, h) the Coriolis term) from initial conditions to T+120h averaged over the 90 cases (a–d) in CNTL, and (e–h) their differences between CNTL and GLN. Note that the colour scales in (a–d) and (e–h) are different.

constituting model physics (i.e., convection, boundary layer, and gravity wave drag) are shown in Fig. 7. A comparison of Fig. 4(a–c) with Fig. 6(a–c) indicates that the resolved processes (Fig. 4(b)) and the residual term (Fig. 4(c)) in the momentum budget correspond to the dynamics tendency (Fig. 6(b)) and the physics parametrizations tendency (Fig. 6(c)) in the model, respectively. Although the residual term is relatively small from the upper troposphere to the stratosphere, there is a recognizable westward forcing around the NH mid-latitude lower stratosphere. The residual term (Fig. 4(c)) near the surface balances with the Coriolis forcing (Fig. 5(d)). A relatively large deceleration by the residual term is shown in the upper stratosphere to the mesosphere (i.e., a deceleration of the westerly wind in winter hemisphere and a deceleration of the easterly wind in summer hemisphere). Most of the residual term can be accounted for by unresolved momentum transports which are being parametrized by the gravity wave drag scheme, the boundary layer scheme, and the convection scheme shown in Fig. 7.

370

375

The quasi-error in the total tendency is dominated by the large compensation between Coriolis forcing (Fig. 5(h)) and the residual term (Fig. 4(f)), particularly in the troposphere and the stratosphere. Whilst the Coriolis term in GLN may have a non-negligible error as described in Sect. 2.6, a verification of zonal-mean meridional wind in CNTL and GLN against the MetUM analysis demonstrates that the error of the Coriolis term in GLN is smaller than that in CNTL (not shown). This

**Figure 6.** Latitude-height cross sections of zonal-mean (a, e) total tendency of zonal wind, and their breakdown into model tendencies of (b, f) dynamics, (c, g) physics, and (d, h) nudging [m s<sup>-1</sup> day<sup>-1</sup>] from initial conditions to T+120h averaged over the 90 cases (a–d) in CNTL, and (e–h) their differences between CNTL and GLN. Zonal wind tendency due to nudging in CNTL is completely null. Note that the colour scales in (a–d) and (e–h) are different.

**Figure 7.** Latitude-height cross sections of a breakdown of the model physics zonal-wind tendency (shown in Fig. 6(c)) to the individual parametrizations, (a) convection, (b) boundary layer, and (c) gravity wave drag [m s<sup>-1</sup> day<sup>-1</sup>] in CNTL from initial conditions to T+120h averaged over the 90 cases.

suggests that the difference of the Coriolis forcing between CNTL and GLN could approximate the error in Coriolis term

**Figure 8.** Latitude-height cross sections of zonal-mean (a, d) total tendency of temperature, and budget contributions from (b, e) the resolved processes (i.e., sum of the mean stationary flow term, the stationary eddy heat flux term, and the transient eddy heat flux term shown in Fig. 9) and (c, f) the residual term of the zonal-mean thermal budget equation [K day<sup>-1</sup>] written in Eq. (3) from initial conditions to T+120h averaged over the 90 cases (a–c) in CNTL, and (d–f) their differences between CNTL and GLN. Contour in (a), for reference, indicates the temperature forecast at T+120h which is identical with that in Fig. 1(d). Note that the colour scales in (a–c) and (d–f) are different.

in CNTL against truth. The westerly wind bias in the midlatitudes over the upper troposphere to the middle stratosphere is attributable to the quasi-error in Coriolis forcing around the tropopause and the residual term in the middle stratosphere, which is examined further in Sect. 5. As the nudging increments are comparable to the forecast errors at least those growing over the 6-hour timescale, a similarity between the quasi-error in the residual term (Fig. 4(f)) and the reversed sign of the nudging forcing in GLN (Fig. 6(h)) is consistent with the finding of the previous studies that fastest growing errors at short timescales are likely associated with the model physics (Martin et al., 2010; Ma et al., 2014; Martin et al., 2021). Little difference in the physics parameterization tendencies between CNTL and GLN, shown in Fig. 6(g), demonstrates that the tendency diagnostics using GLN are less useful to estimate correct unresolved physics tendency to be parameterized in the model and indicates that the physics schemes themselves have substantial deficiencies. The quasi-error in the Coriolis forcing may be a response to the incorrect parametrized mechanical or thermal forcing.

**Figure 9.** Latitude-height cross sections of a breakdown of the resolved processes (shown in Fig. 8(b, e)) to the individual terms of the zonal-mean thermal budget equation [K day $^{-1}$ ] written in Eq. (3) (i.e., from left to right, (a, d) the mean stationary flow term, (b, e) the stationary eddy heat flux term, and (c, f) the transient eddy heat flux term) from initial conditions to T+120h averaged over the 90 cases (a–c) in CNTL, and (d–f) their differences between CNTL and GLN. Note that the colour scales in (a–c) and (d–f) are different.

#### 4.2 Thermal budgets

390

395

400

In the same manner as the zonal momentum budget analysis, the total tendencies of zonal-mean temperature can be diagnosed using the thermal budget analysis as written in Eq. (3). Figures 8 and 9 exhibit latitude-height distributions of the mean total forecast tendency and its thermal budget from initial conditions to Day 5, and their quasi-error against GLN. In CNTL shown in Fig. 9(a–c), the mean stationary flow component has a vertically consistent structure in the troposphere (Fig. 9(a)). Separating it into the meridional advection, the vertical advection, and the adiabatic heating with mean vertical motions reveals that a contribution of the meridional advection is considerably small (not shown). In general, the vertical advection (the second term of the mean stationary flow component in Eq. (3)) and the adiabatic cooling (warming) with ascents (descents) (the third term of the mean stationary flow component in Eq. (3)) compensate throughout the troposphere because the vertical pressure derivative of temperature is positive. As a result, the adiabatic heating with vertical motions dominates over the vertical temperature advection. The mean stationary flow component illustrates the Hadley circulation leading to the cooling with ascents in the SH tropics and the warming with descents in the NH tropics. The cooling by the residual term in the middle to the upper troposphere in the NH high latitude (Fig. 8(c)) counteracts a warming by the transient and stationary eddy heat flux convergence (Fig. 9(b, c)). In the stratospheric winter hemisphere, the mean stationary flow component, dominated by the

**Figure 10.** Latitude-height cross sections of zonal-mean (a, e) total tendency of temperature, and their breakdown into model tendencies of (b, f) dynamics, (c, g) physics, and (d, h) nudging [K day<sup>-1</sup>] from initial conditions to T+120h averaged over the 90 cases (a–d) in CNTL, and (e–h) their differences between CNTL and GLN. Temperature tendency due to nudging in CNTL is completely null. Note that the colour scales in (a–d) and (e–h) are different.

adiabatic heating with vertical motions, and the eddy heat flux component offset each other, and the cooling by the residual term balances with the warming due to the resolved dynamical processes. By contrast, in the stratospheric summer hemisphere where the eddy components are quiescent (Fig. 9(b, c)), the warming by the residual term counterbalances the cooling by the mean flow which constitutes a part of the stratospheric meridional circulations (Figs. 8(c) and 9(a)).

405

410

As with the momentum budget, the residual term in each experiment corresponds to the parametrized forcing shown in Fig. 10(c). Figure 11 presents contributions to the model physics tendency from the individual parametrizations and shows that the cooling and the warming in the stratosphere are attributable to dominant longwave and shortwave radiative processes in winter and summer hemisphere, respectively (Fig. 11(a, b)). The vertical and latitudinal distribution of the residual term is more complex in the troposphere. Latent heat release and subgrid heat transport as well as radiative processes play a substantial role, and these processes interact and balance each other. The warming in the tropics in summer hemisphere appears to correspond with latent heat release and subgrid heat transport represented by model physics (e.g., convection, cloud, microphysics, vertical mixing, etc.). The cooling in the upper troposphere is associated with a dominant longwave radiation process partly compensated by a shortwave radiative heating.

**Figure 11.** Latitude-height cross sections of a breakdown of the model physics temperature tendency (shown in Fig. 10(c)) to the individual parametrizations, (a) shortwave radiation, (b) longwave radiation, (c) convection, (d) microphysics, (e) cloud, (f) boundary layer, and (g) gravity wave drag [K day<sup>-1</sup>] in CNTL from initial conditions to T+120h averaged over the 90 cases.

The quasi-errors in the individual budget terms against GLN, presented in Figs. 8(d-f) and 9(d-f), show that there are large discrepancies in the mean stationary flow component (Fig. 9(d)) and the residual term (Fig. 8(f)) relative to GLN in the troposphere to the middle stratosphere. A set of negative and positive errors around the equator throughout the troposphere suggests a meridional displacement of the large-scale ascents in summer hemisphere. The quasi-error in the residual term shows a qualitatively similar distribution with that in the total tendency especially from the upper troposphere to the middle stratosphere (Fig. 8(d, f)), which is not the case in the zonal momentum budget (Fig. 4(d, f)). This suggests that the unresolved thermodynamic processes are the main sources of the temperature error from the upper troposphere to the middle stratosphere, and an error generated by deficient unresolved processes at a very short-range timescale is not completely compensated for by model dynamics responses even at Day 5. The similarity between the nudging forcing with the reversed sign (Fig. 10(h)) and the quasi-error in the residual term ((Fig. 8(f)) suggests that initial growing error at the very short range timescale, which the nudging would reduce in GLN, is associated with model physics parameterizations. There are likely to be systematic deficiencies in the relevant physics parametrizations (e.g., a deficient cooling or an excessive warming around the tropical tropopause, a deficient warming or an excessive cooling in the middle stratosphere, etc.).

Another interesting characteristic in CNTL is an underestimated warming by the stationary eddy component (mainly the meridional convergence of the stationary eddy heat flux) in the NH high latitude from the upper troposphere to the lower

stratosphere compared with GLN (Fig. 9(b, e)). This indicates that the cold bias in CNTL arises from the resolved dynamical processes as well as from the residual term. The warming due to the eddy heat flux convergence in the lower stratosphere appears to be substantially affected by the sudden stratospheric warming (SSW) event that has occurred in the end of December 2018 (Rao et al., 2019). This is revealed by the timeseries of the thermal budget at a certain pressure level in the stratosphere (for example in Fig. 14).

## 5 NH mid-latitude wind bias

Given the study of the large-scale structure of the model biases and the momentum and thermal budgets, we investigate specific model biases, the NH mid-latitude wind bias and the relevant temperature biases, in detail. To identify sources of the NH mid-latitude stratospheric zonal wind errors and the relevant temperature errors shown in Fig. 2, we analyse the zonal momentum and thermal budgets, as well as the model dynamics, physics and nudging tendencies over the domains A, B, and C (see Sect. 3). Finally, temperature-only nudging experiments provide experimental insights on impacts of thermal forcing on the NH mid-latitude wind bias.

# 5.1 Zonal momentum and thermal budgets

Zonal momentum budgets and model tendencies averaged over the NH midlatitude are examined. Figure 12 shows timeseries of the zonal momentum budget and the model tendencies due to the dynamics, physics, and nudging forcing in the individual forecasts up to Day 1 (from initial conditions to T+24h) and Day 5 (from initial conditions to T+120h) averaged over the NH mid-latitude band (30°N-60°N) at 50 hPa (see the black line A in Fig. 1(e)). The zonal momentum budget and physics parametrization tendencies up to Day 1 as well as Day 5 are shown in Fig. 12 to clarify differences in contributions to the total tendency between Day 1 with initial easterly bias and Day 5 with subsequent westerly bias. Note that the total tendencies depicted in Fig. 12(g-j) are not identical with those depicted in the zonal momentum budget (Fig. 12(c-f)) because the model tendencies at the specific model level closest to the pressure level of 50 hPa are selected whilst the momentum budget is analysed in pressure coordinates. In 1-day CNTL forecasts, the stationary eddy momentum flux and the Coriolis forcing contribute to an acceleration of the westerly wind, and the residual term decelerates and partly offsets the acceleration (Fig. 12(c)). Decelerations by the residual term are quantitatively consistent with those by the physics parametrization tendency dominated by the gravity wave drag (Fig. 12(c, g)). Unlike the consistent deceleration by the residual term in CNTL (Fig. 12(c)), that in GLN shows a temporal variation with accelerations and decelerations day by day (Fig. 12(c, e)). In almost all the cases, whilst the Coriolis term provides excessive westerly tendencies relative to GLN, the residual term has excessive easterly tendencies against GLN (Fig. 12(e)). The other resolved processes including the mean stationary flow advection and the eddy components show smaller quasi-errors against GLN. The error compensation between the Coriolis term and the residual term results in a net westerly quasi-error in CNTL against GLN (Fig. 12(e)), even though CNTL has the easterly wind bias according to the verification against the MetUM analysis (Fig. 12(a)). This suggests that the validation of the budget in CNTL against GLN could mislead the total tendency error in the case that GLN has relatively large errors against analysis in comparison

**Figure 12.** Timeseries area-averaged over the NH mid-latitude  $(30^{\circ}\text{N}-60^{\circ}\text{N})$  at 50 hPa depicted by the line A in Fig. 1(e) of (a, b) zonal wind errors against the MetUM analyses [m s<sup>-1</sup>], (c, d) zonal momentum budget in CNTL [m s<sup>-1</sup> day<sup>-1</sup>], (e, f) differences in the zonal momentum budget (CNTL minus GLN) [m s<sup>-1</sup> day<sup>-1</sup>], (g, h) individual model physics, dynamics, and nudging tendencies in CNTL [m s<sup>-1</sup> day<sup>-1</sup>], and (i, j) differences in the individual model physics, dynamics, and nudging tendencies (CNTL minus GLN) [m s<sup>-1</sup> day<sup>-1</sup>]. Left column shows Day 1 forecasts: (a) the errors at a T+24h forecast lead time, and (c, e, g, i) temporal changes from initial conditions to T+24h. Right column shows Day 5 forecasts: (b) the errors at a T+120h forecast lead time, and (d, f, h, j) temporal changes from initial conditions to T+120h. Note that the nudging tendency in (g, h) is completely null, and physics parametrization tendencies of the convection scheme and the boundary layer scheme are omitted in (g–j) because they are negligibly small in this case.

to CNTL, for example at a short timescale. The quasi-error in the residual term can be accounted for by the nudging forcing (Fig. 12(e, i)), which accelerates the NH mid-latitude lower stratosphere wind in GLN (Fig. 12(i)). It is probable that direct mechanical forcing predominantly from the parametrized gravity wave drag is a source of the easterly wind bias at the very initial stage of the forecasts. On the other hand, the westerly wind errors growing thereafter could be caused by dynamical responses possibly to diabatic heating and temperature biases. The easterly forcing due to the gravity wave drag becomes slightly stronger in GLN (Fig. 12(i)), implying that the scheme responds to the westerly nudging forcing. Another interesting point is that there is little difference in the zonal momentum budget and in the model tendencies between up to Day 1 and up to Day 5. This implies that budget analysis at a short timescale is also beneficial to understand contributions to the total tendency at a medium-range timescale except for the partition among the advective terms even if characteristics of the error are different among timescales. Although the model zonal wind error is positive at lead time Day 5 (Fig. 12(b)), the nudging tendency at that lead time is still positive in GLN (Fig. 12(j)). This is because the nudging forcing effectively corrects the model error against analysis at shorter timescales (depending on the relaxation timescale and nudging frequency) rather than at longer timescales.

To investigate the relevant temperature bias, thermal budgets and model temperature tendencies averaged over the NH tropics around the tropopause are examined. Figure 13 presents timeseries of the thermal budget and the temperature tendencies due to dynamics, physics, and nudging averaged over the NH tropics to subtropics band (0°–30°N) at 70 hPa (see the black line B in Fig. 1(f)). Figure 13(a, b) reveals that CNTL has positive temperature errors at a 1-day forecast lead time in almost all the cases and at a 5-day forecast lead time in all the cases, indicating a robustness of the warm bias around the tropical tropopause. The dominant warming by the residual term leads to positive total tendencies, although the resolved processes contribute to a cooling and partly offset the residual warming (Fig. 13(c, d)). The quasi-errors in the individual terms against GLN show a substantial agreement between the total tendencies and the residual term (Fig. 13(e, f)). This agreement indicates that the residual term plays a key role in the error of the total tendency at both Day 1 and Day 5. The residual term corresponds to the diabatic warming parametrized in the longwave and shortwave radiation schemes (Fig. 13(e, f)) and the tendency due to nudging (Fig. 13(i, j)) demonstrates that the direct nudging forcing accounts for the quasi-error in the residual term. It is likely that deficiencies in the radiation schemes or model variables ingested into the schemes cause the tropical tropopause warm bias, although further investigation is needed.

The temperature bias over the NH high latitude around the tropopause, which is another relevant temperature bias, is also addressed. Figure 14 shows the thermal budget and the model tendencies averaged over the NH high-latitude band (60°N–90°N) at 200 hPa (see the black line C in Fig. 1(f)). Compared with the NH tropics region shown in Fig. 13, the stationary eddy component plays an important role in the budget especially from the beginning of the December to the mid-January (Fig. 14(c, d)). This might respond to a dynamical situation prior to and during the SSW. The warming by the stationary eddy heat flux component is underestimated in CNTL compared with GLN throughout the experimental period (Fig. 14(e, f)). The magnitude of the temperature nudging forcing in this domain (Fig. 14(i, j)) is smaller than that in the previous domain (Fig. 13(i, j)). In contrast to the NH tropics where the residual term is consistently overestimated in CNTL relative to GLN (Fig. 13(e, f)), the residual term shows the positive and negative quasi-error in the NH high latitude (Fig. 14(e, f)). These suggest that the NH

**Figure 13.** Same as Fig. 12, but area-averaged over the NH tropics to subtropics  $(0^{\circ}-30^{\circ}N)$  at 70 hPa depicted by the line B in Fig. 1(f) of (a, b) temperature errors against the MetUM analyses [K], (c, d) thermal budget in CNTL [K day<sup>-1</sup>], (e, f) differences in the thermal budget (CNTL minus GLN) [K day<sup>-1</sup>], (g, h) individual model physics, dynamics, and nudging tendencies in CNTL [K day<sup>-1</sup>], and (i, j) differences in the individual model physics, dynamics, and nudging tendencies (CNTL minus GLN) [K day<sup>-1</sup>]. Note that physics parametrization tendencies of the convection scheme, the cloud scheme, the microphysics scheme, and the gravity wave drag scheme are omitted in (g-j) because they are negligibly small in this case.

high-latitude temperature bias arises from dynamical responses as well as model physics. The initial temperature error in this domain (Fig. 14(a)) is also less robust than in the NH tropics (Fig. 13(a)).

**Figure 14.** Same as Fig. 13, but area-averaged over the NH high-latitude region  $(60^{\circ}\text{N}-90^{\circ}\text{N})$  at 200 hPa depicted by the line C in Fig. 1(f). Note that physics parametrization tendencies of the convection scheme, the cloud scheme, the microphysics scheme, and the gravity wave drag scheme are omitted in (g-j) because they are negligibly small in this case.

Given the diagnostics of the error evolution and the momentum and thermal budget analysis, a possible mechanism of the forecast error growth is summarized. The main source of the NH mid-latitude lower-stratospheric westerly wind bias in CNTL is probably the temperature biases around the tropopause especially in the tropics. The mechanical forcing itself has an erroneous excessive easterly tendency (and/or a deficient westerly forcing) at least at N320 horizontal resolution which causes the NH mid-latitude easterly wind bias at an initial stage of the forecasts. The overly strong temperature gradient causes the

**Figure 15.** Same as Fig. 2(d), but for various nudging experiments summarized in Table 1. CNTL and GLN are identical with those in Fig. 2(d). Dotted lines with open circles and smaller caps of error bars indicate nudging sensitivity experiments in which variables are relaxed only around the tropopause.

subsequently developing westerly wind bias through the excessive eastward Coriolis forcing. There are compensating errors between an excessive westerly tendency caused by thermal forcing originating probably from the radiative processes and an excessive easterly tendency predominantly from the gravity wave drag. Possible deficiencies in the parametrizations in the MetUM need to be investigated further and are discussed in Sect. 6.

# 510 5.2 Nudging sensitivity experiments

515

A partial nudging technique is adopted to attempt to disentangle compensating errors and understand remote impacts of temperature nudging forcing particularly on the NH mid-latitude lower-stratospheric wind bias, which corroborates findings of the possible error mechanism summarized above. The thermal forcing due to the temperature nudging can, in some sense, be interpreted as an additional artificial diabatic heating required to correct model biases. Figure 15 shows time evolutions of NH mid-latitude zonal wind error at 50 hPa in the various nudging experiments outlined by Table 1. Many interesting experimental results are found. The westerly wind bias in CNTL is eliminated by the global temperature nudging, and conversely an easterly wind bias emerges and evolves gradually and monotonically up to Day 15. The temperature error around the tropical tropopause is probably one of the main drivers of the NH mid-latitude wind bias in the lower stratosphere. Note that, however, these experimental results do not determine genuine sources of the wind errors because errors that the nudging directly miti-

**Figure 16.** Latitude-height cross sections of differences in zonal-mean (a1–d1) total tendency of zonal wind, and budget contributions from the individual terms of the zonal-mean zonal momentum budget equation (i.e., from left to right, (a2–d2) the mean stationary flow advection term, (a3–d3) the stationary eddy momentum flux term, (a4–d4) the transient eddy momentum flux term, (a5–d5) the Coriolis term, and (a6–d6) the residual term) [m s<sup>-1</sup> day<sup>-1</sup>] written in Eq. (2) from initial conditions to T+120h averaged over the 90 cases in (a) NHTrpT minus CNTL, (b) NHTrpTrpT minus CNTL, (c) NHHLT minus CNTL, and (d) NHHLTrpT minus CNTL. Green dashed boxes indicate the temperature-nudging domain including interfacing transition zones and layers. The black horizontal line A, the same as that in Fig. 1(e), are inserted for reference. Note that the colour scale is different from that in Fig. 4.

gates are contaminated at a longer forecast lead time by those which are remotely generated by dynamical responses. Impacts of the temperature nudging are significantly different between the whole column nudging and the tropopause nudging over the NH high latitude. There is a difference in the timescales on which temperature nudging over the tropics and at the high latitudes act. The temperature nudging over the tropics reduces substantially the wind bias at shorter timescales, whereas the impact of temperature nudging over the high latitudes is relatively substantial at Day 9 and afterwards. This suggests a difference in the timescales of the temperature error growth in CNTL in these regions. The temperature error around the tropical tropopause evolves more quickly than that in the NH high-latitude tropopause (cf. Figs. 13(a) and 14(a)), and therefore the impact of nudging also appears more rapidly.

Figure 17. Same as Fig. 16, but for momentum budgets from initial conditions to T+240h.

Zonal momentum budget analysis could reveal the contribution from the individual terms and help understand the differences in impact timescales of the temperature nudging over the tropics and the high latitudes. Changes in the momentum budget of each experiment relative to that of CNTL are presented for Day 5 and Day 10 in Fig. 16 and Fig. 17, respectively. Day 5 represents a case on which the impact on the wind error by tropical-tropopause temperature nudging is dominant, while Day 10 is a case on which the effects by high-latitude temperature nudging are more significant. Impacts outside the nudged domain shown by green dashed boxes are interpreted as a remote response to errors in the target nudged domain. Momentum budgets up to Day 5 show that changes of the total tendencies from CNTL in the NH mid-latitude lower stratosphere are dominated by changes of the Coriolis forcing in all the temperature nudging experiments shown in Fig. 16. Temperature nudging alters the meridional circulation through geopotential gradients, which is clearly demonstrated by a meridional momentum budget analysis (not shown), and consequently the Coriolis forcing in the zonal momentum budgets (Fig. 16(a5-d5)). The stationary eddy momentum flux convergence also contributes to the deceleration on the poleward side of where the Coriolis forcing decelerates the westerly wind (Fig. 16(a3-d3)). In NHTrpT and NHTrpTrpT, the residual term also has a substantial contribution to the deceleration in the NH midlatitude (Fig. 16(a6, b6)). Note that temperature is relaxed towards the analysis, but zonal wind is not relaxed at all in the temperature nudging experiments. Therefore, the zonal wind tendencies due to nudging are

completely zero, and the difference in the residual term of zonal momentum budgets is attributed to the effects other than the nudging. Hence, a different responses of parameterizations to temperature nudged grid-box mean fields or an imbalance caused by the temperature nudging may lead to the deceleration in the residual term. In the momentum budgets up to Day 10 in the NH tropics temperature nudging experiments shown in Fig. 17(a, b), the eddy momentum flux convergence and the Coriolis forcing weakens the total deceleration in the lower and higher latitude domain, respectively, in the NH lower stratosphere (Fig. 17(a3-a5, b3-b5)).

There is a difference in the impacts of nudging between the NH tropics nudging and the NH high-latitude nudging with respect to a latitudinal spreading of the impacts and their vertical distribution. The temperature nudging over the NH tropics affects not only the tropics but also the mid- to high-latitude regions, and its impact gradually extends from the low-latitude region to the high-latitudes in both hemispheres. On the other hand, the impact of temperature nudging over the NH high latitude is confined to the extratropics. The NH tropics nudging shows vertically tripolar structure in the total tendency around the tropics. The total deceleration in the lower stratosphere is confined to below the middle stratosphere and overlaid by acceleration in the upper stratosphere. On the other hand, the NH high-latitude nudging decelerates zonal wind consistently from the upper troposphere to the stratosphere. This might also be associated with the different vertical structure of the temperature error between the tropics and the high latitudes shown in Fig. 1(b, d, f).

#### 6 Conclusions

To understand mechanisms of model systematic errors in the global MetUM and identify their sources, we have applied the momentum and thermal budget analysis and the nudging technique to NWP simulations in boreal winter. The model systematic errors addressed in this paper include a westerly wind bias in the NH midlatitude from the upper troposphere to the stratosphere, a warm temperature bias around the tropical tropopause, and a cold temperature bias around the tropopause over the poles. The warm and cold temperature errors near the tropopause develop quickly and systematically in almost all the cases, indicating their robustness. On the other hand, the NH mid-latitude wind in the lower stratosphere has an initial easterly bias especially over the Himalayas and the Tibetan Plateau and a subsequent westerly bias in the zonal-mean zonal wind emerging at Day 3 and evolving thereafter. The momentum and thermal budget analysis is a practical diagnostic method that specifies to what extent the individual components in the budget equation contribute to the total tendency of wind or temperature. The residual term corresponds to the effects of unresolved processes parametrized in GCMs on the total tendency. Simulations in which global horizontal wind and temperature are relaxed towards the MetUM analysis can be considered as a proxy of the analysis with some caveats, and their momentum and thermal budgets are used as the reference budgets to evaluate "quasi-error" in the budget components. The zonal momentum budget analysis reveals that the westerly wind bias in the midlatitudes over the upper troposphere to the middle stratosphere is attributable to the dominant quasi-error in the Coriolis forcing around the tropopause and the quasi-error in the residual term over the middle stratosphere. It is probable that an excessive mechanical forcing due to the orographically induced gravity wave drag is one of the sources of the easterly wind bias at the very initial stage of the forecasts. On the other hand, with respect to the thermal budget, the quasi-error in the residual term dominates that in the total 575 temperature tendency from the upper troposphere to the middle stratosphere. Hence, the temperature bias is probably caused by systematic deficiencies in the unresolved processes represented by physics parametrizations. The NH high-latitude cold bias appears to be associated with the resolved dynamical processes as well as unresolved physical processes. This mechanism suggests error compensations in the zonal momentum budget.

The momentum and thermal budgets analysis to examine forecast and analysis time evolutions in NWP timescales is expected to be a powerful diagnostic tool not only for identifing sources of model systematic errors but also for understanding impacts of scientific changes or artificial forcing (e.g., nudging) among experiments. From zonal-mean perspective, one possible mechanism of the wind forecast error growth is that an overly strong temperature gradient with the warm (and cold) bias developing quickly near the tropopause leads to an acceleration of the southerly wind through an increased geopotential gradient, consequently resulting in an acceleration of the westerly wind by the Coriolis forcing. The fact that temperature nudging over the NH tropical tropopause can significantly reduce the NH mid-latitude stratospheric westerly wind bias supports the idea that the tropical tropopause warm bias is one of the main sources of the mid-latitude wind bias. The excessive deceleration due to the gravity wave drag should partly compensate the westerly wind errors developing at the longer forecast lead time. Although this paper focuses on the zonal-mean budgets, it will also be profitable to analyse the longitudinally varying budgets (e.g., Yang et al., 2013) because mean total tendencies closely linked to model systematic errors are zonally asymmetric and strongly localized. It is also important to explore a sensitivity of the budgets and model tendencies to the model horizontal resolution. Whilst the Eulerian framework is used in this study, the quasi-Lagrangian framework, such as the Transformed Eulerian Mean (TEM; Andrews and Mcintyre, 1976) or the Mass-weighted Isentropic zonal Means (MIM; e.g., Iwasaki, 1989, 1992; Tanaka et al., 2004) equation, could also be useful for understanding large-scale atmospheric circulations.

Momentum and thermal budgets in the global nudging experiment used in this study should be interpreted cautiously because of errors even in the directly constrained variables and possible underestimations of the eddy components. These errors could be avoided or reduced using the budgets in analysis or reanalysis data instead of the global nudging experiment data as in main previous studies. However, it is required that the temporal, horizontal, and vertical resolutions of the analysis data are identical with those of the forecast data to compare their budgets fairly. It could be quite challenging to assess the budgets in reanalyses and validate the budgets in models because of different resolutions or interpolations to get the resolutions aligned. One promising way for operational meteorological centres to avoid this issue is applying the budget analysis with a different computation and/or definition of temporal mean to an objective analysis created by data assimilation system. If the budget components calculated using the fields at each model time step are well reproduced using 6-hourly instantaneous fields, the analysis data potentially provide a better reference than the global nudged simulation. Defining temporal mean differently as the average over the experimental period, a series of analysis and first guess, which is a 6-hour forecast from the previous objective analysis, is available for diagnosing the budgets in the so-called analysis world and forecast world, although this is for the purpose of identifying sources of initial model errors rather than understanding model error growth. This should also allow us to reduce an initial spinup generated by discrepancies in the configuration between data assimilation systems and forecast models.

We now discuss possible causes of the biases presented above at the physical process level and potential further investigations. The excessively strong deceleration in the NH midlatitude in winter is probably attributed to the orographic gravity wave drag scheme. The vertical integrated momentum budget analysis (Milton and Wilson, 1996) or the angular momentum budget analysis (Brown, 2004; van Niekerk et al., 2016) may help to identify if the gravity wave excitation process or the vertical propagation and deposition process needs to be modified. Since there might be deficiencies in parametrizations employed by the MetUM with specific horizontal resolutions, a horizontal resolution sensitivity also needs to be investigated. The effects of the gravity wave drag scheme on atmospheric circulations depend on the seasons and the hemispheres because of different atmospheric stability and wind fields near the land surface and the asymmetric orography between the hemispheres. Therefore, investigations of model systematic errors for the seasons (e.g., boreal summer) could be informative. With respect to the warm bias around the tropical tropopause, it is possible that there are deficiencies in the radiation schemes or relevant variables including cloud fields. This is essentially consistent with the conclusion of the previous studies (e.g., Hardiman et al., 2015). Further investigations using the diagnostics framework in this study combined with other diagnostic methods are a promising approach to establishing how to improve the model representations.

Code and data availability. Due to intellectual property right restrictions, we cannot provide either the source code or the documentation papers for the Met Office Unified Model (MetUM). The MetUM is available for use under licence. For further information on how to apply for a licence, see https://www.metoffice.gov.uk/research/approach/collaboration/unified-model/partnership (Met Office, 2024). JULES is available under licence free of charge. For further information on how to gain permission to use JULES for research purposes, see https://jules.jchmr.org/ (JULES, 2024). Model data used in this study are archived at the Met Office and are available to research collaborators upon request.

Author contributions. CM conducted the experiments, performed the analyses, and prepared the paper with contributions from all authors. SFM initiated the study, designed the experiments and the analyses, and gave important guidance and feedback. JMR designed and conducted the experiments, and gave important guidance and feedback.

Competing interests. The contact author has declared that none of the authors has any competing interests.

Acknowledgements. This study was carried out in a collaboration between the Met Office and Japan Meteorological Agency while C. Matsukawa was visiting the Met Office. J. M. Rodríguez was funded by the Met Office Climate Science for Service Partnership (CSSP) China project under the International Science Partnerships Fund (ISPF). Insightful and supportive comments from two anonymous reviewers are gratefully acknowledged.

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
