# Peer review of "Process-based diagnostics using atmospheric budget analysis and nudging technique to identify sources of model systematic errors"

_EGUsphere, 2025_

## Author Comment (AC1)

Response to the comments from referee on "Process-based diagnostics using atmospheric budget analysis and nudging technique to identify sources of model systematic errors" by C. Matsukawa et al.

We thank the referee #1 for their valuable comments and constructive suggestions for improving the paper. We will make extensive corrections to a revised manuscript. The comments by referee #1 are listed below in black italics, followed by our responses in blue.

Response to Referee #1

**Under the category "General Comments"**

*This paper addresses the interesting questions regarding the development of model errors in temperature and zonal wind in the Met Office UM, using a nudging relaxation and a tendency budget decomposition. These methods are well described, and used to come to some interesting results. In the most part the paper is well written, the results well presented and the conclusions good, however I would suggest a substantial re-write of Section 4 and additional reference to related work before the paper is accepted for publication.*

Thank you for your suggestive comments. We agree that some descriptions of atmospheric budget diagnostics, particularly an interpretation of the residual term, are confusing and a substantial re-write is needed for improving the manuscript based on your comments and our responses to them below. Your suggestions of some additional references are very informative. We will cite more related work in a revised manuscript.

*In several places the text would be easier to follow if each panel were given unique identifying letters. Figures 2 and 12 are good, figures 3-11 would benefit from additional labels.*

We agree to your suggestive comments. Figures 1, 3, 4, 5, 6, 7, 8, 9, 10, 11, 16, and 17 will be replotted with given unique identifying letters in each panel.

**Under the category "Major comments"**

*L228-238, L430-432: As this is not the first article to identify temperature and wind biases in models, it would be appropriate to provide in the introduction at least a brief review or acknowledgment of the previous work on model temperature and wind biases (particularly the lower stratosphere zonal wind, and tropical tropopause temperature) (in addition to the brief mention of two examples on L555 at the end) and some summary of the existing*

*knowledge. By providing a sufficient overview of the related work that has already been published you will be able to make it much clearer to readers the original contribution that this article makes in the context of the current state of knowledge.*

In the conclusion part of the manuscript, the two papers, Hardiman et al. (2015) and Bland et al. (2021), have been being cited to discuss further studies. However, as the referee #1 suggested, it would be better if these papers could be moved to Section 1 "Introduction" and additional papers could be cited there as an example of previous studies.

*As I understand it, the residual term for CNTL is the sum of the parameterised processes, any numerical integration errors and differences arising from the primitive budget equations being different from the actual equations of the UM. The CNTL-GLN difference in the residual is the difference of these in the CNTL and GLN plus the nudging, which is the very short timescale model error in the UM. However there is no single, early, concise explanation of this.*

As the referee #1 mentioned, the residual term in our study is interpreted to be qualitatively equivalent to estimated unresolved forcing and contain the sum of the parameterized processes, any numerical integration errors, any budget diagnostics calculation errors (e.g., conversion from the model levels to the pressure levels, second-order centered difference scheme for meridional and vertical derivatives calculation, etc.), and differences of the governing equations of the UM and the primitive budget equations. However, we would like to emphasize that the model tendencies diagnostics (i.e., Figs. 6, 7, 10, 11, 12(g-j), 13(g-j), 14(g-j)) and atmospheric budget diagnostics (i.e., Figs. 4, 5, 8, 9, 12(c-f), 13(c-f), 14(c-f) are completely independent in their calculations. Model tendencies are calculated in the model during the time integration at each model level using dynamical core and physics parameterizations. On the other hand, atmospheric budgets used in our study are evaluated from forecast data (specifically U, V, T, and Omega fields) on pressure levels as shown by the budget equations (Eq. (2) and (3)) after the completion of the time integration. The atmospheric budgets calculation doesn't require the model tendencies calculated by dynamical core and physics parameterizations at all. That's why we think that the atmospheric budget diagnostics can be compared with the model tendencies to demonstrate that the resolved term and the unresolved term in the atmospheric budget equation are qualitatively equivalent to the model tendencies of dynamics and physics parameterizations respectively (as shown by a comparison between Fig. 4(a) and Fig. 6(a) and between Fig. 8(a) and Fig. 10(a)). As you mentioned, the discrepancy between the governing equations of the UM (nonhydrostatic, fully compressible deep-atmosphere equations of motion) and the primitive equations could account for a difference between the model tendencies and atmospheric budget diagnostics.

We agree to your comment and a single, early, concise explanation for this will be added to the manuscript. We will create Section 2.5 and 2.6 newly and move some explanations in Section 4 to explain how these two independent diagnostics link and the difference in budgets/tendencies between CNTL and GLN can be interpreted in a revised manuscript.

*While various aspects of the above are written in various places in various ways, references to components of this and interpretations of the residual often seem confusing. For example lines 302-304 present the similarity as if it were not true by definition. The article could benefit from a concise re-write of lines 327-335 to clearly explain the interpretation of the residual, and moved earlier in the section. Then any subsequent discussion of the residual or any of its components can be re-considered in the context of the earlier explanation to make the section overall more cohesive and consistent.*

As we mentioned above, the model tendencies diagnostics and atmospheric budget diagnostics are completely independent. Therefore, the similarity between Fig. 4 and Fig. 6 is not necessarily true by definition and is worth being demonstrated. We agree that the earlier explanation of independency between the model tendencies diagnostics and the atmospheric budget diagnostics will improve this paper. We will create Section 2.5 and 2.6 newly to explain how these two diagnostics link in a revised manuscript.

*A consequence of this is: taking the mean CNTL-GLN tendency from dynamics (4b, 6b, 8b, 10b), will this not be disproportionately dominated by the model dynamics bias in the first 6 hours being balanced by the nudging, therefore masking any information about differences between CNTL and GLN at timescales longer than this? If this is what you are trying to address in lines 320-324 it is not clear. With terms being so heavily dominated by the nudging, what information can be gotten from the dynamics/residual at timescales of longer than 24 hours in any quantities including data from GLN?*

In the atmospheric budget analysis shown in Fig. 4 and Fig. 8, we use GLN with 6-hour relaxation timescale to make quasi-analysis data and use as a best estimate of truth of atmospheric budgets and verify the budgets in CNTL against GLN. In the nudging experiments, forced variables (i.e., U, V, and T) are relaxed towards the MetUM analysis data every model time step throughout the integration up to 15 days in this study. Therefore, the

GLN has small error against MetUM analysis as shown in Fig. 2(d) even at longer forecast lead time than 24 hours. This is the reason we use the atmospheric budgets in GLN as a best estimate of truth (GLN is equivalent to a data assimilation/analysis as used in this study).

The choice of the relaxation timescale of nudging experiments is arbitrary. The shorter the relaxation timescale is, the stronger the model prognostic variables are relaxed towards forcing data (i.e. MetUM analysis data) throughout the model integration. As described in the cited previous study (Telford et al. 2008), too long relaxation timescale is ineffective to use a nudging experiment as a best estimate of truth, but too short relaxation timescale nudging could make the model unstable.

Differences in the atmospheric budgets between CNTL and GLN shown in Figs 4(b) and 8(b) exhibit the error in the atmospheric budgets of CNTL against GLN. On the other hand, the model tendencies shown in Fig. 6(b) and Fig. 10(b) are just the difference between experiments and doesn't show any error in the model dynamics or model physics parameterizations at all.

*It is not clear that figure 4 provides any useful information not contained within figures 5 and 6 (and similarly that figure 8 does not show anything not in figures 9-11) other than the contour overlay which could be moved. It may be possible to make this section more concise by removing figures 4 & 8.*

We feel that Fig. 4 does provide key information on overall model systematic bias in zonal wind whereas Fig. 5 provides the breakdown of that error into budget components coming from different aspects of the general circulation in the model (CNTL) and analysis (GLN). Figure 6 shows the equivalent breakdown from the tendency diagnostics (on model levels) which brings in the role of parametrized sub-grid scale physical forcing (gravity wave drag etc.) alongside the resolved circulation or Dynamics (made up of the mean flow, stationary & transient eddies, and Coriolis terms shown in Fig. 5.). Similar arguments apply for Fig. 8 with respect to Figs. 9 and 11.

To provide a little more detail , calculations of the model tendencies diagnostics (i.e., Figs. 6, 7, 10, 11, 12(g-j), 13(g-j), 14(g-j)) and atmospheric budget diagnostics (i.e., Figs. 4, 5, 8, 9, 12(c-f), 13(c-f), 14(c-f) are independent. The atmospheric budgets diagnostics don't require the model tendencies calculated by dynamical core and physics parameterizations during the model integration. That's why we think that the atmospheric budget diagnostics (Fig. 4) can be evaluated against the model tendencies (Fig. 6) to demonstrate that the resolved term and the unresolved term in the atmospheric budget equation (Fig. 4) are qualitatively equivalent

to the model tendencies of model dynamics and physics parameterizations (Fig. 6) respectively. Figs 4 and 8 (zonal momentum budget and thermal budget) are necessary to compare with Figs 6 and 10 (model zonal wind tendency and temperature tendency).

Figure 5 shows a breakdown of the resolved processes which consist of mean stationary flow component, stationary eddy component, transient eddy component, and Coriolis forcing as shown by Eq. (2). Therefore, the resolved processes shown in the middle panel of Fig. 4 is completely equal to the sum of the 4 components shown in Fig. 5 (as mentioned in the caption of Fig. 4). Similarly, Fig. 7 shows a breakdown of the physics parameterization tendency which consist of convection, boundary layer, and gravity wave drag. Therefore, physics parameterization tendency shown in the third column is completely equal to the sum of the 3 parameterization tendencies shown in Fig. 7.

Similarly, Figs 8-11 show the comparison between two independent diagnostics and their breakdown for temperature.

*L323: This information not shown would perhaps benefit from being shown. With such an emphasis on the Coriolis term in the zonal wind budget, the article would benefit from the inclusion of further discussion of any biases in the meridional wind field.*

Thank you so much for your informative comments.

An example of meridional wind error of CNTL and GLN against analysis is shown below:

[Figure]

Figure A1 (Left) Zonal-mean meridional wind of CNTL and MetUM analysis, and error in meridional wind of CNTL against analysis from Day 1 to Day 5, and (Right) zonal-mean meridional wind of GLN and MetUM analysis, and error in meridional wind of GLN against analysis from Day 1 to Day 5. As can be seen from Fig. A1, the GLN has a much smaller error in meridional wind than CNTL.

We have evaluated the meridional momentum budget equation (not shown in this paper) that can be derived in the same manner as the zonal momentum budget equation (Eq. (2)). These

additional budget diagnostics are expected to provide useful information on meridional wind fields. The reason why the meridional momentum budget diagnostics have been omitted from our paper is that they are less important than zonal momentum and thermal budgets in the context of this study because northward geopotential gradient term and southward Coriolis forcing term are almost balanced (i.e. geostrophic balance) and other terms including temporal derivative term, momentum flux convergence term, and residual term are relatively small in the meridional momentum budget.

*L330-331: Is this conclusion necessarily true? Unsure if the results provided support this.*

Error of a given variable against analysis can be written below:

$$
\begin{aligned}
& x_e(t_i; \Delta t) \\
& \equiv x_f(t_i; \Delta t) - x_a(t_i + \Delta t) \\
& = \{x_f(t_i; \Delta t) - x_f(t_i; 0)\} - \{x_a(t_i + \Delta t) - x_a(t_i)\} \\
& = \left\{ \overline{\left(\frac{\partial x}{\partial t}\right)_{f,t_i}} - \overline{\left(\frac{\partial x}{\partial t}\right)_{a,t_i}} \right\} \Delta t
\end{aligned}
$$

where $x_e(t_i; \Delta t)$ is an error at a forecast lead time of $\Delta t$ initialized at $t_i$, $x_f(t_i; \Delta t)$ is a forecast at a lead time of $\Delta t$ initialized at $t_i$, $x_a(t_i + \Delta t)$ is an analysis at $t_i + \Delta t$. This equation would be helpful and we will include it in Section 2.6 newly added.

Similarity between residual term error and total tendency error, which is proportional to the error of the corresponding variable itself as shown in the equation described above, suggests that the error of the corresponding variable itself stems from the residual term error. As shown by comparison of Fig. 4(a-3) with Fig. 6(a-3), the residual term in the budget equation is qualitatively equivalent to the physics parameterization representing unresolved processes in the model. These results suggest the conclusion described in L330-331.

*L367-370: The sign of the CNTL-GLN difference in physics is the opposite of the sign of the error in 10-b-3 and 10-b-1, which seems to suggest the opposite to what this sentence is saying.*

We understand that you have questioned the opposite sign of the difference between two experiments shown in Figs. 10-b-3 and 10-b-1 particularly in the upper stratosphere above 35km altitude. Figures 4(b), 5(b), 8(b), and 9(b) show differences in individual components of the atmospheric budget equations between CNTL and GLN, which can be interpreted as an error of the individual budget components against GLN, the best estimate of truth, in this study. On the other hand, Figures 6(b) and 10(b) show differences in model tendencies between CNTL and GLN. For instance, a difference in physics parameterization tendencies

is a difference in physics parameterization responses to CNTL and GLN grid-box mean fields given to the parameterization calculations. Physics parameterization scheme itself has deficiencies and that used in CNTL and GLN is identical. Figure 10(b)3 shows a difference in the response of physics parameterizations rather than error in physics parameterization tendencies. L367-370 doesn't describe this difference in physics parameterization tendencies.

*L323-324: By "the error in Coriolis term against truth" do you mean the error in the Coriolis term in CNTL against truth? If so, state this.*

As indicated by the referee #1, this sentence mentions "the error in the Coriolis term in CNTL against truth". We will revise this sentence.

**Under the category "Additional specific comments"**
*All figures: It would benefit the reader if each individual panel were assigned a letter identifier (a), (b), etc.*

We agree to your suggestive comments. Figures 1, 3, 4, 5, 6, 7, 8, 9, 10, 11, 16, and 17 will be replotted with given unique identifying letters in each panel.

*L33-34: There have been a wide variety of diagnostic methods… It would be appropriate to list some.*

Thank you so much for your suggestive comments.
The diagnostic methods and their activities to evaluate model systematic errors to be listed include, for instance, single-column models experiments (e.g., Duynkerke et al. 2004; Lenderink et al. 2004; Svensson et al. 2011), model intercomparison projects (e.g., Zadra et al. 2013; van Niekerk et al. 2020, Elvidge et al. 2019), WGNE conferences on model systematic errors (Zadra et al. 2018; Frassoni et al. 2023), potential vorticity budget diagnostics (e.g., Chagnon et al. 2013; Saffin et al. 2016), semi-geostrophic balance tool (Sánchez et al. 2020), perturbed parameter ensemble technique (Sexton et al. 2019; Karmalkar et al. 2019; Williams et al. 2020). We will cite some of these papers in a revised manuscript.
- Duynkerke et al. 2004: Observations and numerical simulations of the diurnal cycle of the EUROCS stratocumulus case, QJRMS, https://doi.org/10.1256/qj.03.139
- Lenderink et al. 2004: The diurnal cycle of shallow cumulus clouds over land: A single-column model intercomparison study, QJRMS, https://doi.org/10.1256/qj.03.122

- Svensson et al. 2011: Evaluation of the Diurnal Cycle in the Atmospheric Boundary Layer Over Land as Represented by a Variety of Single-Column Models: The Second GABLS Experiment, *Boundary-Layer Meteorol*., https://doi.org/10.1007/s10546-011-9611-7
- Zadra et al. 2013: WGNE drag project, https://collaboration.cmc.ec.gc.ca/science/rpn/drag_project/
- van Niekerk et al. 2020: COnstraining ORographic Drag Effects (COORDE): A Model Comparison of Resolved and Parametrized Orographic Drag, JAMES, https://doi.org/10.1029/2020MS002160
- Elvidge et al. 2019: Uncertainty in the Representation of Orography in Weather and Climate Models and Implications for Parameterized Drag, https://doi.org/10.1029/2019MS001661
- Zadra et al. 2018: Systematic Errors in Weather and Climate Models: Nature, Origins, and Ways Forward, BAMS, https://doi.org/10.1175/BAMS-D-17-0287.1
- Frassoni et al. 2023: Systematic Errors in Weather and Climate Models: Challenges and Opportunities in Complex Coupled Modeling Systems, https://doi.org/10.1175/BAMS-D-23-0102.1
- Chagnon et al. 2013: Diabatic processes modifying potential vorticity in a North Atlantic cyclone, QJRMS, https://doi.org/10.1002/qj.2037
- Saffin et al. 2016: The non-conservation of potential vorticity by a dynamical core compared with the effects of parametrized physical processes, QJRMS, https://doi.org/10.1002/qj.2729
- Sánchez et al. (2020): Linking rapid forecast error growth to diabatic processes, https://doi.org/10.1002/qj.3861
- Sexton et al. (2019): Finding plausible and diverse variants of a climate model. Part 1: establishing the relationship between errors at weather and climate time scales, https://doi.org/10.1007/s00382-019-04625-3
- Karmalkar et al. (2019): Finding plausible and diverse variants of a climate model. Part II: development and validation of methodology, https://doi.org/10.1007/s00382-019-04617-3
- Williams et al. (2020): Addressing the causes of large-scale circulation error in the Met Office Unified Model, QJRMS, https://doi.org/10.1002/qj.3807

*L124-128: I can discern what you mean, but it would be beneficial to re-write these sentences to make them clearer.*

Thank you so much for your comments.

These sentences describing the nudging increments or nudging tendencies will be revised to make them clearer.

*L191: This is not a general property of all partial differential equations, so remove these opening 4 words (or re-phrase the sentence to make it more accurate).*

In this paragraph, we would like to explain a difference in the budget equations between the partial differential equations themselves shown in Eq. (2, 3) and discretized version of the equations. Equations considered in the numerical model or the diagnosed forecast/analysis data is the discretized one. As you mentioned, this is not a general property of all partial differential equations. We think that the phrases "*In the partial differential equations of Eqs. (2) and (3),* " and "*the residual term diagnosed using spatially discretized data*" are clearer and will revise the manuscript so.

*L253-254: I do not see evidence for this statement. Provide evidence, or clarify that this is not shown.*

The sentence in L253-254 "*The rapid error growth and consistent drifts suggest that these errors are relevant to the model physics rather than changes in the atmospheric circulation.*" is based on the suggestion by previous work (e.g. Martin et al., 2010; Ma et al., 2014; Martin et al., 2021) cited in the Introduction part. We will cite these papers again in L253-254.

*L262-263 & Figures 2d, 3b, 12a, 12b: The independence of GLN state with forecast lead time seems strange. Is the GLN state at lead times >24 hours truly identical? Can you explain this or comment further on why the differences are zero or imperceptibly small?*

The GLN at forecast lead times except for 0 hours should be truly identical.
In the GLN, forced variables (i.e., U, V, and T) are relaxed towards the MetUM analysis data every model time step throughout the integration up to 15 days. Mean error can be formulated as follows

$$\text{ME} = \sum_{i=1}^{N} \phi_e(t_i; \Delta t) = \sum_{i=1}^{N} \{\phi_f(t_i; \Delta t) - \phi_a(t_i + \Delta t)\}$$

where $\phi_e(t_i; \Delta t)$ is an error at a forecast lead time of $\Delta t$ initialized at $t_i$, $\phi_f(t_i; \Delta t)$ is a forecast at a lead time of $\Delta t$ initialized at $t_i$, $\phi_a(t_i + \Delta t)$ is an analysis at $t_i + \Delta t$, and $N$ is the number of the cases. For example, mean zonal wind error at 1-day forecast lead time is calculated from 90 cases initialized between at 30[th] Nov. 2018 and at 27[th] Feb. 2019, and mean

zonal wind error in 15-day forecast lead time is calculated from 90 cases initialized between at 16th Nov. 2018 and 13th Feb. 2019. Regardless of forecast lead time, mean zonal wind error is calculated for the fixed validity period from 1st Dec. 2018 to 28th Feb. 2019. It is obvious that the MetUM analysis used as the forcing data doesn't depend on forecast lead time. Therefore, model prognostic variables forced in nudging experiments are relaxed towards the same analyzed fields regardless of forecast lead time, which results in the same forecast fields at specific validity time. This can account for the reason why the forced variables in the nudging experiments have an error independent on forecast lead time.

*L292-293: a decrease in the tropics... State, a decrease in what in the tropics. Similarly for lines 293-295.*

Thank you for your suggestive comment. We will revise the manuscript as below:

L292: a decrease in zonal-mean zonal wind in the tropics with ...

L293: a decrease in zonal-mean zonal wind in the mid-latitude with ...

L294: an increase in zonal-mean zonal wind in the subtropic with ...

*Figure 12-14: Are the differences between the dynamics and the sum of the resolved flow components, and the differences between the residual tern and the sum of the nudging + physics, in panels f compared to panels j, larger than one would expect resulting from the model level v.s. pressure level comparison? They seem quite large by eye. Can you comment on this please?*

In Figs 12-14, panels (f) show error in atmospheric budget components of CNTL against GLN, and panels (j) show differences in model tendencies between CNTL and GLN. As we mentioned above, a similarity between them is not necessarily true by definition even on the same vertical coordinate (i.e., given model level v.s. the model level, or given pressure level v.s. the pressure level). In addition to this, we compare the atmospheric budgets on given pressure levels (i.e., 50 hPa in Fig 12, 70 hPa in Fig 13, and 200 hPa in Fig. 14) in panels (c-f) and model tendencies on the specific model levels closest to the respective pressure levels. Thus, the larger difference between panels (f) and panels (j) than one would expect results from not only the difference of their vertical coordinate but also the 2 independent diagnostics which are not necessarily equivalent by definition.

*L479-480: This result seems important but it is not shown in this article (nor do you state in the text that it is not shown). With such an emphasis on the Coriolis term in the zonal wind*

*budget, the article would benefit from the inclusion of further discussion of any biases in the meridional wind field.*

A word "(not shown)" will be put in to just after "which is clearly demonstrated by a meridional momentum budget analysis".
The Coriolis term in the zonal momentum budget equation, $fv$, is proportional to meridional wind speed, $v$. Therefore, the change in zonal-mean Coriolis term between specific two experiments shown in Fig. 16, $f\bar{v}_{\mathrm{EXP}} - f\bar{v}_{\mathrm{CNTL}} = f(\bar{v}_{\mathrm{EXP}} - \bar{v}_{\mathrm{CNTL}})$, although a latitudinal dependency of Coriolis parameter $f = 2\Omega \sin\phi$ needs to be considered. As shown in Figure A1 in this response, CNTL has a southerly wind bias in NH subtropics in the lower stratosphere. As can be seen in Fig. 5(b)4 and Fig. 16(a)5, the error in Coriolis term against GLN can be reduced by temperature nudging, which implies changes in meridional wind.
We will add further discussion of biases in the meridional wind fields related to model biases of zonal wind and temperature addressed in this study.

*L483-485: Returning to earlier comments on the interpretations of the residual term – this idea would benefit from further explanation and possibly inclusion of the "not shown" material.*

As we mentioned above, calculations of the model tendencies and atmospheric budgets are completely independent. We suppose that the earlier explanation rewritten in a revised manuscript could be beneficial. In the NHTrpT and NHTrpTrpT, temperature is relaxed towards the analysis but zonal wind is not relaxed at all. Therefore, zonal wind tendency due to nudging in the NHTrpT and NHTrpTrpT is completely zero. The sentence "The difference in the residual term of NHTrpT and NHTrpTrpT cannot be fully accounted for by changes in the physics parametrization tendencies (not shown)" describes a discrepancy of the difference in the residual term and the difference in the parameterization tendencies.
We will add some additional explanations above to a revised manuscript and rewrite this sentence.

**Under the category "Technical Corrections"**
*L29: within the first few days*
*L133-134: 10 degrees in the horizontal ⋯ two model levels in the vertical*
*L136: from December 2018 to February 2019 (or from the December of 2018 to the February of 2019)*
*L206: the word 'definitely' doesn't need to be here*
*L242: we focus on three⋯*

*L262: but do not depend⋯*

*L538: "specially" is the wrong word to use here. One way to rewrite this would be for example "one promising way for operational ⋯"*

These technical corrections are really helpful, and all of them will be reflected to the revised manuscript.

---

## Author Comment (AC2)

Response to comments from referees on "Process-based diagnostics using atmospheric budget analysis and nudging technique to identify sources of model systematic errors" by C. Matsukawa et al.

We thank the referee for the valuable comments and constructive suggestions.
We have listed the comments by referee #2 below in black italics, followed by our responses below in blue.

Reply to Referee #2

*The reviewer would like to thank the authors for their comprehensive study on model-based diagnostics to identify sources of model systematic errors, which could be useful for any NWP and climate models. The method successfully detects sources and origins of model forecast errors in the Met Office Unified Model, which could be common in the state-of-the-art operational and research atmospheric GCMs. I think The paper is comprehensive and well organized as a WCD paper, so that I would already recommend minor revision, though I have several comments below before acceptance.*

Thank you so much for your reviewing and suggestive comments. We respond to your specific comments below.

**Under the category "Specific comments"**

*1) The relaxation timescale of 6 hours could be practically useful to reduce/erase the initial errors that can grow up in subsequent forecasts. I am curious that how sensitive is "tau" (timescale) to your conclusion? For example, if you choose 12 or 24 hours as tau, you could obtain consistent results, that is, could detect the same origins causing the NH wind field errors?*
*The relaxation timescale might be similar with the evaluation time of the Forecast Sensitivity to Observations (Hotta et al. 2017, Prive et al. 2020). The shorter timescales might be more practical for forecasts.*
*(Hotta et al. 2017, MWR, DOI:10.1175/MWR-D-16-0290.1; Prive et al. 0220, QJRMS, DOI:10.1002/qj.3909)*

We use the GLN experiment to make quasi-analysis data and use as a best estimate of truth of atmospheric budgets. In the nudging experiments, forced variables (i.e., U, V, and T) are relaxed towards the MetUM analysis data every model time step throughout the integration

up to 15 days in this study. The relaxation timescale "tau" determines how strong the model prognostic variables are relaxed towards forcing data rather than how long the nudging is applied to. Therefore, the GLN has small error against MetUM analysis as shown in Fig. 2(d) even at longer forecast lead time than 6 hours.

The choice of the relaxation timescale of nudging experiments is arbitrary. The shorter the relaxation timescale is, the stronger the model prognostic variables are relaxed towards forcing data (i.e. MetUM analysis data) throughout the model integration. As described in the cited previous study (Telford et al. 2008), too long relaxation timescale is ineffective to use a nudging experiment as a best estimate of truth, but too short relaxation timescale nudging could make the model unstable. We will add an additional explanation mentioned above to Section 2.2.

This paper doesn't focus on a difference in the relaxation timescale, but it sounds interesting and might provide an interesting result which could be useful for understanding a different timescale of error contribution of each component in the budget equations.

*2) Could you comment in your conclusion part about two topics that could be useful as future studies?*

Thank you so much for your suggestive comments.
These specific comments are really helpful to improve our manuscript and our response to two comments kindly given us are described below.

*- Budget analysis based on conservation values, potential vorticity or angular momentum (TEM or MIM), could be useful for your further budget analysis. Currently I only refer to a paper by Kobayashi and Iwasaki (2016), though there may be other important studies to be cited.*
*(Kobayashi and Iwasaki 2016, JGR-A, DOI:10.1002/2015JD023476)*

Thank you so much for your informative suggestions.
We will cite potential vorticity and angular momentum equations in the conclusion part.

*- As a comparison with the experiment for 2018 boreal winter, additional experiment for an SH winter could be helpful to improve your conclusion (e.g., Yamazaki et al. 2023).*
*(Yamazaki et al. 2023, WAF, DOI:10.1175/WAF-D-22-0159.1)*

We agree to your suggestive comments. Some conclusions in our study depend on seasons

and hemispheres. For example, gravity wave drag, which could be one of the sources of the wind error, is expected to be more active in NH because of the orographic features and also in winter because of the wind fields and atmospheric stabilities near land surface. Even if the temperature bias around tropical tropopause wouldn't depend on season, balanced situation of the compensating errors would change and result in different consequences. We are wondering that a focus on specific season and hemisphere makes this paper detailed and less confusing. However, we will implement some comments described above into Section 6 in a revised manuscript. This could help to improve our conclusion and discussion.

---

## Author Response (AR1)

The comments by referee #1 and referee #2 are listed below in black italics, followed by our point-by-point responses in blue and relevant author's changes in the manuscript in green. Note that a specific number of lines in green in this document is applicable to the revised manuscript with track changes.

**Response to comments by Referee #1**

**Under the category "General Comments"**

This paper addresses the interesting questions regarding the development of model errors in temperature and zonal wind in the Met Office UM, using a nudging relaxation and a tendency budget decomposition. These methods are well described, and used to come to some interesting results. In the most part the paper is well written, the results well presented and the conclusions good, however I would suggest a substantial re-write of Section 4 and additional reference to related work before the paper is accepted for publication.

Thank you for your suggestive comments. We agree that some descriptions of atmospheric budget diagnostics, particularly an interpretation of the residual term, are confusing and a substantial re-write is needed for improving the manuscript. Your suggestions of some additional references are very informative. As responded below in "Major comments" section in detail, new sections (Section 2.5 and 2.6) have been created in earlier part of the revised manuscript for describing the detail of a correspondence between model tendencies and budget diagnostics, and a validation of budget components against the global nudging experiment. Some of the descriptions in Section 4 and other parts of the manuscript have been moved to the new sections.

In several places the text would be easier to follow if each panel were given unique identifying letters. Figures 2 and 12 are good, figures 3-11 would benefit from additional labels.

We agree to your suggestive comments. As described below in "Additional specific comments" section in detail, Figures 1, 3, 4, 5, 6, 7, 8, 9, 10, 11, 16, and 17 have been replotted with given unique identifying letters in each panel. Additionally, based on the suggestive comments from editors, the color schemes in some of the figures (i.e., Figs 2, 12, 13, 14, 15) have been revised for readers with color vision deficiencies.

**Under the category "Major comments"**

L228-238, L430-432: As this is not the first article to identify temperature and wind biases in

models, it would be appropriate to provide in the introduction at least a brief review or acknowledgment of the previous work on model temperature and wind biases (particularly the lower stratosphere zonal wind, and tropical tropopause temperature) (in addition to the brief mention of two examples on L555 at the end) and some summary of the existing knowledge. By providing a sufficient overview of the related work that has already been published you will be able to make it much clearer to readers the original contribution that this article makes in the context of the current state of knowledge.

In the conclusion part of the manuscript, the two papers, Hardiman et al. (2015) and Bland et al. (2021), have been being cited to discuss further studies. However, as the referee #1 suggested, it would be better if these papers could be moved to Section 1 "Introduction". L68-L73, L652-L656: The two papers which are dedicated to investigating temperature biases in the MetUM, Hardiman et al. (2015) and Bland et al. (2021), have been moved from the Conclusion part to the Introduction part and summarized.

As I understand it, the residual term for CNTL is the sum of the parameterised processes, any numerical integration errors and differences arising from the primitive budget equations being different from the actual equations of the UM. The CNTL-GLN difference in the residual is the difference of these in the CNTL and GLN plus the nudging, which is the very short timescale model error in the UM. However there is no single, early, concise explanation of this.

While various aspects of the above are written in various places in various ways, references to components of this and interpretations of the residual often seem confusing. For example lines 302-304 present the similarity as if it were not true by definition. The article could benefit from a concise re-write of lines 327-335 to clearly explain the interpretation of the residual, and moved earlier in the section. Then any subsequent discussion of the residual or any of its components can be re-considered in the context of the earlier explanation to make the section overall more cohesive and consistent.

As the referee #1 mentioned, the residual term in our study is interpreted to be qualitatively equivalent to estimated unresolved forcing and contains the sum of the parameterized processes, any numerical integration errors, any budget diagnostics calculation errors (e.g., conversion from the model levels to the pressure levels, second-order centered difference scheme for meridional and vertical derivatives calculation, etc.), and differences of the governing equations of the UM and the primitive budget equations. Model tendencies are calculated in the model online during the time integration at each model level using dynamical

core and physics parameterizations. On the other hand, atmospheric budgets used in our study are evaluated from forecast data (specifically U, V, T, and Omega fields) on pressure levels as shown by the budget equations (Eq. (2) and (3)) offline after the completion of the time integration. The discrepancy between the governing equations of the UM (non-hydrostatic, fully compressible deep-atmosphere equations of motion) and the primitive equations could account for a difference between the model tendencies and atmospheric budget diagnostics.

The follow-up comment by referee #1 in terms of the tendency diagnostics is essentially correct, although budgets can be calculated using every model time step fields, since mean fields denoted by overbars, for instance  $[\overline{u'v'}] = [\overline{u}\overline{v} - \overline{u}\overline{v}]$  and  $[\overline{u}^*\overline{v}^*] = [\overline{u}\overline{v}] - [\overline{u}][\overline{v}]$ , are derived using every model time step fields.

In contrast to our study, previous studies (e.g., Milton and Wilson (1996)) present the budget residual using a different definition that directly refers to the model physics tendencies. Since the budget calculations in this study do not refer to model tendencies explicitly, we have mentioned that they are considered to be independent, to emphasize the difference in the definition compared to previous studies. However, they are implicitly linked through their governing equations, with different assumption and approximation as indicated by referee #1. We think it is inappropriate to consider the two diagnostics to be completely independent.

A single, early, concise explanation for this have been added to the manuscript. We have created two new sections, 2.5 and 2.6, to explain how these two diagnostics are linked and the difference in budgets/tendencies between CNTL and GLN. We have moved some explanations in Section 4 and other parts of the manuscript to bring together similar descriptions into one part, Section 2.5 and 2.6.

L246-L266: Section 2.5 has been added to explain interpretations of atmospheric budgets and model tendencies, and importance of the comparison to demonstrate their quantitative correspondence.

L267-L301: Section 2.6 has been added to describe how we validate the atmospheric budgets in CNTL against GLN to decompose an error in the total tendencies into individual budget components. For ease of understanding, we refer to a difference in budget components between CNTL and GLN as "quasi-error" in the revised manuscript.

L167-L168, L222-L223, L356-L358, L381-L382, L392-L400, L409-L413: The descriptions on the diagnostics that had sat in Section 4 and other parts have been moved to Section 2.5 and 2.6 for earlier explanations.

L358, L365, L401, L407, L414-L415, L417, L422-L423, L447, L451, L486, L487, L491, L512, L517, L528, L601-L602, L603, L604, L606-L607: We refer to a difference in budget components between CNTL and GLN as "quasi-error" in the revised manuscript.

A consequence of this is: taking the mean CNTL-GLN tendency from dynamics (4b, 6b, 8b, 10b), will this not be disproportionately dominated by the model dynamics bias in the first 6 hours being balanced by the nudging, therefore masking any information about differences between CNTL and GLN at timescales longer than this? If this is what you are trying to address in lines 320-324 it is not clear. With terms being so heavily dominated by the nudging, what information can be gotten from the dynamics/residual at timescales of longer than 24 hours in any quantities including data from GLN?

In the atmospheric budget analysis shown in Fig. 4 and Fig. 8, we use GLN with 6-hour relaxation timescale to make quasi-analysis data and use as a best estimate of truth of atmospheric budgets and verify the budgets in CNTL against GLN. In the nudging experiments, forced variables (i.e., U, V, and T) are relaxed towards the MetUM analysis data every model time step throughout the integration up to 15 days in this study. Therefore, the GLN has small error against MetUM analysis as shown in Fig. 2(d) even at longer forecast lead time than 24 hours. This is the reason we use the atmospheric budgets in GLN as a best estimate of truth (GLN is equivalent to a data assimilation/analysis as used in this study).

The choice of the relaxation timescale of nudging experiments is arbitrary. The shorter the relaxation timescale is, the more strongly the model prognostic variables are relaxed towards forcing data (i.e. MetUM analysis data) throughout the model integration. As described in the cited previous study (Telford et al. 2008), too long relaxation timescale is ineffective to use a nudging experiment as a best estimate of truth, but too short relaxation timescale nudging could make the model unstable.

Differences in the atmospheric budgets between CNTL and GLN shown in Figs  $4\frac{(b)}{(d-f)}$  and  $8\frac{(b)}{(d-f)}$  exhibit the error in the atmospheric budgets of CNTL against GLN. On the other hand, the model tendencies shown in Fig.  $6\frac{(b)}{(e-h)}$  and Fig.  $10\frac{(b)}{(e-h)}$  are just the difference between experiments and doesn't show any error in the model dynamics or model physics parameterizations at all.

L132-L136: In order to explain how the nudging works and why the 6-hour relaxation timescale was selected, we have added some explanations described above.

L168-L169: The reason why relatively short relaxation time scale is used in the nudging experiment in this study has been described.

L296-L301: The last paragraph in Section 2.6 explains interpretations of differences in model tendencies between CNTL and GLN, which is a difference in the response to the different grid-box mean fields rather than error in the model tendencies.

L392-L400, L403: Explanations of non-negligible error in GLN budgets against truth have

**been moved to Section 2.6.**

It is not clear that figure 4 provides any useful information not contained within figures 5 and 6 (and similarly that figure 8 does not show anything not in figures 9-11) other than the contour overlay which could be moved. It may be possible to make this section more concise by removing figures 4 & 8.

We feel that Fig. 4 does provide key information on overall model systematic bias in zonal wind whereas Fig. 5 provides the breakdown of that error into budget components coming from different aspects of the general circulation in the model (CNTL) and analysis (GLN). Figure 6 shows the equivalent breakdown from the tendency diagnostics (on model levels) which brings in the role of parametrized sub-grid scale physical forcing (gravity wave drag etc.) alongside the resolved circulation or Dynamics (made up of the mean flow, stationary & transient eddies, and Coriolis terms shown in Fig. 5.). Similar arguments apply for Fig. 8 with respect to Figs. 9 and 11. We think Figs. 4 and 8 are essential for conclusion of this study in order to demonstrate a correspondence between budget diagnostics shown in Figs. 4(a-c) and 8(a-c) and model tendencies shown in Figs. 6(a-c) and 10(a-c) and exhibit a quasi-error in resolved processes and unresolved processes against GLN shown in Figs. 4(d-f) and 8(d-f). Figures 4-11: Their captions have been revised for clarification.

L323: This information not shown would perhaps benefit from being shown. With such an emphasis on the Coriolis term in the zonal wind budget, the article would benefit from the inclusion of further discussion of any biases in the meridional wind field.

Thank you so much for your informative comments.

An example of meridional wind error of CNTL and GLN against analysis is shown below:

Figure A1 (Left) Zonal-mean meridional wind of CNTL and MetUM analysis, and error in

meridional wind of CNTL against analysis from Day 1 to Day 5, and (Right) zonal-mean meridional wind of GLN and MetUM analysis, and error in meridional wind of GLN against analysis from Day 1 to Day 5. As can be seen from Fig. A1, the GLN has a much smaller error in meridional wind than CNTL.

We have evaluated the meridional momentum budget equation (not shown in this paper) that can be derived in the same manner as the zonal momentum budget equation (Eq. (2)). These additional budget diagnostics are expected to provide useful information on meridional wind fields. The reason why the meridional momentum budget diagnostics have been omitted from our paper is that they are less important than zonal momentum and thermal budgets in the context of this study. In the meridional momentum budget, northward geopotential gradient term and southward Coriolis forcing term are almost balanced (i.e. geostrophic balance) and other terms including temporal derivative term, momentum flux convergence term, and residual term are less dominant in the meridional momentum budget.

L330-331: Is this conclusion necessarily true? Unsure if the results provided support this.

Error of a given variable against analysis can be written below:

$$x_{e}(t_{i}; \Delta t)$$

$$\equiv x_{f}(t_{i}; \Delta t) - x_{a}(t_{i} + \Delta t)$$

$$= \left\{x_{f}(t_{i}; \Delta t) - x_{f}(t_{i}; 0)\right\} - \left\{x_{a}(t_{i} + \Delta t) - x_{a}(t_{i})\right\}$$

$$= \left\{\left(\frac{\partial x}{\partial t}\right)_{f,t_{i}} - \left(\frac{\partial x}{\partial t}\right)_{a,t_{i}}\right\} \Delta t$$

where  $x_e(t_i; \Delta t)$  is an error at a forecast lead time of  $\Delta t$  initialized at  $t_i$ ,  $x_f(t_i; \Delta t)$  is a forecast at a lead time of  $\Delta t$  initialized at  $t_i$ ,  $x_a(t_i + \Delta t)$  is an analysis at  $t_i + \Delta t$ .

Similarity between residual term error and total tendency error, which is proportional to the error of the corresponding variable itself as shown in the equation described above, suggests that the error of the corresponding variable itself stems from the residual term error. As shown by comparison of Fig. 4(a-3)(c) with Fig. 6(a-3)(c), the residual term in the budget equation is qualitatively equivalent to the physics parameterization representing unresolved processes in the model. These results suggest the conclusion described in L330-331412-413.

L268-L278: We have included the equation described above in Section 2.6 newly added for ease of understanding of relation between model errors and model tendency errors.

L409-L413: The descriptions on the interpretation of the residual term have been moved to Section 2.5 and 2.6 and written for earlier concise explanations.

L367-370: The sign of the CNTL-GLN difference in physics is the opposite of the sign of the

error in 10-b-3 and 10-b-1, which seems to suggest the opposite to what this sentence is saying.

We understand that you have questioned the opposite sign of the difference between two experiments shown in Figs. 10-b-310(g) and 10-b-110(e) particularly in the upper stratosphere above 35km altitude. Figures 4(b)4(d-f), 5(b)(e-h), 8(b)(d-f), and 9(b)(d-f) show differences in individual components of the atmospheric budget equations between CNTL and GLN, which can be interpreted as an error of the individual budget components against GLN, the best estimate of truth in this study. On the other hand, Figures 6(b)(e-h) and 10(b)(e-h) show differences in model tendencies between CNTL and GLN. For instance, a difference in physics parameterization tendencies is a difference in physics parameterization responses to CNTL and GLN grid-box mean fields given to the parameterization schemes. Physics parameterization scheme itself has deficiencies and that used in CNTL and GLN is identical. Figure 10(b)3(g) shows a difference in the response of physics parameterizations rather than error in physics parameterization tendencies. L367-370L452-455 is not intended to describe this difference in physics parameterization tendencies.

L296-L301: It has been explained that the difference in physics parameterization tendencies between CNTL and GLN is a difference in physics parameterization responses to CNTL and GLN grid-box mean fields given to the same parameterization schemes.

L323-324: By "the error in Coriolis term against truth" do you mean the error in the Coriolis term in CNTL against truth? If so, state this.

As indicated by the referee #1, this sentence mentions "the error in the Coriolis term in CNTL against truth".

L406: We have revised this sentence.

**Under the category "Additional specific comments"**

All figures: It would benefit the reader if each individual panel were assigned a letter identifier (a), (b), etc.

We agree to your suggestive comments. Figures 1, 3, 4, 5, 6, 7, 8, 9, 10, 11, 16, and 17 have been replotted with given unique identifying letters in each panel. Additionally, based on the suggestive comments from editors, the color schemes in some of the figures (i.e., Figs 2, 12, 13, 14, 15) have been revised for readers with color vision deficiencies.

Figure 1 has subtitles with given unique identifying letters (a-f) for each panel with their explanations in the caption.

Figure 2 has been changed to a new color scheme using the Coblis, as editors provided suggestive comments.

Figure 3 has subtitles with given unique identifying letters (a-d) for each panel with their explanations in the caption.

Figure 4 has subtitles with given unique identifying letters (a-f) for each panel with their explanations in the caption.

Figure 5 has subtitles with given unique identifying letters (a-h) for each panel with their explanations in the caption.

Figure 6 has subtitles with given unique identifying letters (a-h) for each panel with their explanations in the caption.

Figure 7 has subtitles with given unique identifying letters (a-c) for each panel with their explanations in the caption.

Figure 8 has subtitles with given unique identifying letters (a-f) for each panel with their explanations in the caption.

Figure 9 has subtitles with given unique identifying letters (a-f) for each panel with their explanations in the caption.

Figure 10 has subtitles with given unique identifying letters (a-h) for each panel with their explanations in the caption.

Figure 11 has subtitles with given unique identifying letters (a-g) for each panel with their explanations in the caption.

Figure 12 has been changed to a new color scheme using the Coblis, as editors provided suggestive comments.

Figure 13 has been changed to a new color scheme using the Coblis, as editors provided suggestive comments.

Figure 14 has been changed to a new color scheme using the Coblis, as editors provided suggestive comments.

Figure 15 has been changed to a new color scheme using the Coblis, as editors provided suggestive comments.

Figure 16 has subtitles with given unique identifying letters (a1-d6) for each panel with their explanations in the caption.

Figure 17 has subtitles with given unique identifying letters (a1-d6) for each panel.

L321, L347, L350, L369, L371, L379, L383, L384, L385, L387, L402, L415, L423, L424, L432, L435, L436-L437, L441, L447, L448, L452, L458, L474, L508, L522, L567, L569, L570, L578, L587: Changes due to a reference to the panels.

L33-34: There have been a wide variety of diagnostic methods... It would be appropriate to

list some.

**Thank you so much for your suggestive comments.**

L32-L37: Some papers have been cited as an example of the diagnostic methods in the revised manuscript including single-column models experiments (e.g., Duynkerke et al. 2004; Lenderink et al. 2004; Svensson et al. 2011), model intercomparison projects (e.g., van Niekerk et al. 2020, Elvidge et al. 2019), WGNE conferences on model systematic errors (Frassoni et al. 2023), potential vorticity budget diagnostics (e.g., Chagnon et al. 2013; Saffin et al. 2016), semi-geostrophic balance tool (Sánchez et al. 2020), perturbed parameter ensemble technique (Sexton et al. 2019; Karmalkar et al. 2019; Williams et al. 2020).

L124-128: I can discern what you mean, but it would be beneficial to re-write these sentences to make them clearer.

Thank you so much for your comments. The sentences in L<del>124</del>138-L<del>128</del>142 have been rewritten in the revised manuscript.

L138-L142: An explanation of the nudging increments or nudging tendencies has been revised to make it clearer.

L413-L414: A similar description here has been revised as well.

L191: This is not a general property of all partial differential equations, so remove these opening 4 words (or re-phrase the sentence to make it more accurate).

In this paragraph, we would like to explain a difference in the budget equations between the partial differential equations themselves shown in Eq. (2, 3) and discretized version of the equations. Equations considered in the numerical model or the diagnosed forecast/analysis data are the discretized ones. As you mentioned, this is not a general property of all partial differential equations.

L208-L211: The sentences have been written for clarification.

L253-254: I do not see evidence for this statement. Provide evidence, or clarify that this is not shown.

The sentence in L253328-254329 "The rapid error growth and consistent drifts suggest that these errors are relevant to the model physics rather than changes in the atmospheric circulation." is based on the suggestion by previous work (e.g. Martin et al., 2010; Ma et al.,

2014; Martin et al., 2021) cited in the Introduction part.

L330: We have cited the previous studies (e.g. Martin et al., 2010; Ma et al., 2014; Martin et al., 2021) to support this suggestion.

L262-263 & Figures 2d, 3b, 12a, 12b: The independence of GLN state with forecast lead time seems strange. Is the GLN state at lead times >24 hours truly identical? Can you explain this or comment further on why the differences are zero or imperceptibly small?

The GLN at forecast lead times except for 0 hours should be truly identical.

In the GLN, forced variables (i.e., U, V, and T) are relaxed towards the MetUM analysis data every model time step throughout the integration up to 15 days. Mean error can be formulated as follows

$$ME = \sum_{i=1}^{N} \phi_e(t_i; \Delta t) = \sum_{i=1}^{N} \{\phi_f(t_i; \Delta t) - \phi_a(t_i + \Delta t)\}$$

where  $\phi_e(t_i; \Delta t)$  is an error at a forecast lead time of  $\Delta t$  initialized at  $t_i$ ,  $\phi_f(t_i; \Delta t)$  is a forecast at a lead time of  $\Delta t$  initialized at  $t_i$ ,  $\phi_a(t_i + \Delta t)$  is an analysis at  $t_i + \Delta t$ , and N is the number of the cases. For example, mean zonal wind error at 1-day forecast lead time is calculated from 90 cases initialized between at 30th Nov. 2018 and at 27th Feb. 2019, and mean zonal wind error in 15-day forecast lead time is calculated from 90 cases initialized between at 16th Nov. 2018 and 13th Feb. 2019. Regardless of forecast lead time, mean zonal wind error is calculated for the fixed validity period from 1st Dec. 2018 to 28th Feb. 2019. It is obvious that the MetUM analysis used as the forcing data doesn't depend on forecast lead time. Therefore, model prognostic variables forced in nudging experiments are relaxed towards the same analyzed fields regardless of forecast lead time, which results in the same forecast fields at specific validity time. This can account for the reason why the forced variables in the nudging experiments have an error independent of forecast lead time.

L340-L343: Additional concise explanations of consistent error in GLN regardless of forecast lead time have been added.

L292-293: a decrease in the tropics... State, a decrease in what in the tropics. Similarly for lines 293-295.

Thank you for your suggestive comment.

We have revised the manuscript as below:

L371: a decrease in zonal-mean zonal wind in the tropics with ...

L372-L373: a decrease in zonal-mean zonal wind in the mid-latitude with ...

**L374: an increase in zonal-mean zonal wind in the subtropic with ...**

Figure 12-14: Are the differences between the dynamics and the sum of the resolved flow components, and the differences between the residual term and the sum of the nudging + physics, in panels f compared to panels j, larger than one would expect resulting from the model level v.s. pressure level comparison? They seem quite large by eye. Can you comment on this please?

In Figs 12-14, panels (f) show error in atmospheric budget components of CNTL against GLN, and panels (j) show differences in model tendencies between CNTL and GLN. As we mentioned above, a similarity between them is not necessarily true by definition even on the same vertical coordinate (i.e., given model level v.s. the model level, or given pressure level v.s. the pressure level). In addition to this, we compare the atmospheric budgets on given pressure levels (i.e., 50 hPa in Fig 12, 70 hPa in Fig 13, and 200 hPa in Fig. 14) in panels (c-f) and model tendencies on the specific model levels closest to the respective pressure levels. Thus, the larger difference between panels (f) and panels (j) than one would expect results from not only the difference of their vertical coordinate but also the two different diagnostics which are not necessarily equivalent by definition due to the different governing equations and the diagnostics errors.

L479-480: This result seems important but it is not shown in this article (nor do you state in the text that it is not shown). With such an emphasis on the Coriolis term in the zonal wind budget, the article would benefit from the inclusion of further discussion of any biases in the meridional wind field.

The Coriolis term in the zonal momentum budget equation, fv, is proportional to meridional wind speed, v. Therefore, the change in zonal-mean Coriolis term between specific two experiments shown in Fig. 16,  $f\bar{v}_{\text{EXP}} - f\bar{v}_{\text{CNTL}} = f(\bar{v}_{\text{EXP}} - \bar{v}_{\text{CNTL}})$ , is proportional to the impact of nudging on the meridional wind fields, although a latitudinal dependency of Coriolis parameter  $f = 2\Omega \sin \phi$  needs to be considered. As shown in Figure A1 in this response, CNTL has a southerly wind bias in NH subtropics in the lower stratosphere. As can be seen in Fig. 5(b)4(h) and Fig. 16(a)5(a5), the error in Coriolis term against GLN can be reduced by temperature nudging, which implies changes in meridional wind. We would rather put the focus on the zonal momentum budgets and implications of meridional wind from Coriolis term diagnostics in this paper.

L567: A word "(not shown)" has been put in to just after "which is clearly demonstrated by a

meridional momentum budget analysis".

L483-485: Returning to earlier comments on the interpretations of the residual term – this idea would benefit from further explanation and possibly inclusion of the "not shown" material.

We hope that the earlier explanation rewritten in Section 2.5 and 2.6 in the revised manuscript could be beneficial. In the NHTrpT and NHTrpTrpT, temperature is relaxed towards the analysis but zonal wind is not relaxed at all. Therefore, zonal wind tendency due to nudging in the NHTrpT and NHTrpTrpT is completely zero. The sentence "The difference in the residual term of NHTrpT and NHTrpTrpT cannot be fully accounted for by changes in the physics parametrization tendencies (not shown)" describes a discrepancy of the difference in the residual term and the difference in the parameterization tendencies.

L570-L574: This sentence has been rewritten for clearer explanations.

**Under the category "Technical Corrections"**

L29: within the first few days

L133-134: 10 degrees in the horizontal ··· two model levels in the vertical

L136: from December 2018 to February 2019 (or from the December of 2018 to the February of 2019)

L206: the word 'definitely' doesn't need to be here

L242: we focus on three…

L262: but do not depend…

L538: "specially" is the wrong word to use here. One way to rewrite this would be for example "one promising way for operational ···"

**These technical corrections are really helpful.**

All of the technical corrections have been reflected to the revised manuscript as follows:

L29: within the first few days

L149: 10 degrees in the horizontal ··· two model levels in the vertical

L152: from the December 2018 to the February 2019

L225: definitely

L317: we focus on three

L337: but does do not depend

L633: One promising way specially infor operational

Reply to the comments by Referee #2

The reviewer would like to thank the authors for their comprehensive study on model-based diagnostics to identify sources of model systematic errors, which could be useful for any NWP and climate models. The method successfully detects sources and origins of model forecast errors in the Met Office Unified Model, which could be common in the state-of-the-art operational and research atmospheric GCMs. I think The paper is comprehensive and well organized as a WCD paper, so that I would already recommend minor revision, though I have several comments below before acceptance.

Thank you so much for your reviewing and suggestive comments. We respond to your specific comments below.

**Under the category "Specific comments"**

1) The relaxation timescale of 6 hours could be practically useful to reduce/erase the initial errors that can grow up in subsequent forecasts. I am curious that how sensitive is "tau" (timescale) to your conclusion? For example, if you choose 12 or 24 hours as tau, you could obtain consistent results, that is, could detect the same origins causing the NH wind field errors?

The relaxation timescale might be similar with the evaluation time of the Forecast Sensitivity to Observations (Hotta et al. 2017, Prive et al. 2020). The shorter timescales might be more practical for forecasts.

(Hotta et al. 2017, MWR, DOI:10.1175/MWR-D-16-0290.1; Prive et al. 0220, QJRMS, DOI:10.1002/qj.3909)

We use the GLN experiment to make quasi-analysis data and use it as a best estimate of truth of atmospheric budgets. In the nudging experiments, forced variables (i.e., U, V, and T) are relaxed towards the MetUM analysis data every model time step throughout the integration up to 15 days in this study. The relaxation timescale "tau" determines how strongly the model prognostic variables are relaxed towards forcing data rather than how long the nudging is applied to. Therefore, the GLN has small error against MetUM analysis as shown in Fig. 2(d) even at longer forecast lead time than 6 hours.

The choice of the relaxation timescale of nudging experiments is arbitrary. The shorter the relaxation timescale is, the more strongly the model prognostic variables are relaxed towards forcing data (i.e. MetUM analysis data) throughout the model integration. As described in the cited previous study (Telford et al. 2008), too long relaxation timescale is ineffective to use a nudging experiment as a best estimate of truth, but too short relaxation timescale nudging could make the model unstable.

This paper doesn't focus on a difference in the relaxation timescale, but it sounds interesting and might provide an interesting result which could be useful for understanding different timescales of error contribution from each component in the budget equations.

L132-L136: In order to explain how the nudging works and why the 6-hour relaxation timescale was selected, we have added some explanations described above.

L168-L169: The reason why relatively short relaxation time scale is used in the nudging experiment in this study is described.

2) Could you comment in your conclusion part about two topics that could be useful as future studies?

Thank you so much for your suggestive comments.

These specific comments are really helpful to improve our manuscript and our response to the two comments kindly given to us are described below.

- Budget analysis based on conservation values, potential vorticity or angular momentum (TEM or MIM), could be useful for your further budget analysis. Currently I only refer to a paper by Kobayashi and Iwasaki (2016), though there may be other important studies to be cited.

(Kobayashi and Iwasaki 2016, JGR-A, DOI:10.1002/2015JD023476)

**Thank you so much for your informative suggestions.**

L32-L37, L623-L626: We have cited papers on potential vorticity budget analysis (Chagnon et al. 2013; Saffin et al. 2016) in the Introduction part and angular momentum equations (TEM: Andrews and Mcintyre, 1976; MIM: e.g., Iwasaki, 1989, 1992; Tanaka et al., 2004) in the conclusion part.

- As a comparison with the experiment for 2018 boreal winter, additional experiment for an SH winter could be helpful to improve your conclusion (e.g., Yamazaki et al. 2023). (Yamazaki et al. 2023, WAF, DOI:10.1175/WAF-D-22-0159.1)

We agree to your suggestive comments. Some conclusions in our study depend on seasons and hemispheres. For example, gravity wave drag, which could be one of the sources of the wind error, is expected to be more active in NH because of the orographic features and also in winter because of the wind fields and atmospheric stabilities near land surface. Even if the temperature bias around tropical tropopause wouldn't depend on season, balanced situation

of the compensating errors would change and result in different consequences. We are wondering if a focus on specific season and hemisphere makes this paper detailed and less confusing.

L647-L650: We have implemented some comments described above into the Conclusion part in a revised manuscript. We hope this could help to improve our conclusion and discussion.

**Changes to the manuscript not mentioned above**

L126-L128: We have changed the notation for the time interval of the model integration because of the distinction from the forecast length denoted by  $\Delta t$  in L273-L275. The time interval of the model integration adopted in this study has been clarified in L128.

L179: Rewritten for clarification

L202-L203: Rewritten for clarification

L337-L340: Two sentences describing potential errors even in the forced variables in nudging experiments have been moved to Section 2.6 newly created.

L353: Technical correction

L354: Technical correction

L360: Rewritten for clarification

L377-L379: This sentence is not necessary for our conclusion, so it has been deleted for ease of reading

L380: Rewritten for improvements

L382-L383: Parameterizations affecting sub-grid momentum transport has been listed for clarification.

L389-L390: Rewritten for improvements

L405-L406: Rewritten for ease of reading, associated with the "quasi-error" of CNTL against GLN

L476-L479: Rewritten for clarification of explanations of each panel in Figure 12.

L491: Rewritten for clarification

L500-L504: Two sentences have been reordered for ease of reading.

L519: Technical correction

L553, L554: Rephrased for improvements

L610: For clarity, error compensation in *what* is described.

L619: Rewritten for ease of reading

L642-L643: A revision has been added to make the meaning of the word "physical" clearer.

---

## Referee Report (RR1)

Thank you for reading and taking on my comments. I am glad you found them useful, and appreciate the improved clarity of the article. It is good to see that the majority of points raised by both reviewers have been addressed, however, unless I have missed something, it still seems that there is some inconsistency with regards to the comparison between the nudging, the dynamics quasi-error, and the total tendency quasi-error.

For brevity I will not reproduce previous responses in full here, but rather just section headings and opening words.

**Under the category "Major comments"**

As I understand it, the residual term for CNTL...

... However, they are implicitly linked through their governing equations, with different assumption and approximation as indicated by referee #1. We think it is inappropriate to consider the two diagnostics to be completely independent.

...

Thank you for clarifying the agreement on the points that the tendencies and budgets are linked due to the governing equations, and that it is inappropriate to consider them independent. This did not come across in the original text and is useful to have clarified.

The only further note I have on section 2.5 is for line 259, I might suggest changing "but is not necessarily true by definition" to "but they are not necessarily exactly equivalent".

I assume based on lines 260-266 that this is the intended meaning, but as written I do not think the sentence means that. As written is seems something of a contradiction, as is the derivation of the budget not also the definition of the components?

A consequence of this is: taking the mean CNTL-GLN tendency from dynamics (4b, 6b, 8b, 10b), will this not be disproportionately dominated by the model dynamics bias in the first 6 hours being balanced by the nudging...

It may be the case that I was very unclear in my first review, and phrased this badly.

I do not think it is adequately addressed as to why the CNTL-GLN nudging in figure 6(h) appears to have a similar character but with opposite sign to the dynamics in figure 6(f), and whether this is important to the conclusions.

The similarity is noted in line 402, but in lines 413-414 you write that nudging increments are comparable to forecast errors. The forecast error in figure 1(c, e) has a similar character to the total tendency error in 4(d) and 6(e), however these do not have

similar character to the resolved processes error (4e), the Coriolis error (5h) or the dynamics error (6f), which all instead are comparable to the nudging error (6h). Therefore it does not seem that nudging increments are comparable to forecast erros.

It is acknowledged in line 450 that the thermal budget and the momentum budgets are different. Here it is noted that there is similarity between the residual (8f) and the total (8d). This similarity is used to conclude deficiencies in physics parametrizations from the thermal budget. However by using the phrase "unlike the momentum budget", this highlights my confusion that you also conclude for the momentum budget that it is a deficiency in the physics parametrizations in lines 413-417, with similar phrasing, despite the opposite result.

I interpret line 415-417 as saying that the model physics should be applying the same forcing that the nudging is, because you are using a nudged forecast as the truth.

A first look at figure 6 had me confused, as it seemed that the majority of the forcing of the nudging to move the GLN field towards the analysis was being directly un-done and opposed by the tendencies of the model dynamics. The majority of the nudging tendency being directly counteracted seemed to be an odd result, but as I have not personally used nudging in the same way as you are here in this study, it would be very useful if you could please provide your interpretation of what the cancellation of terms in figure 6(f) and 6(h) means, in terms of what the nudging is doing, how the model is responding, and how the differences between CNTL and GLN should be interpreted in light of this.

I feel like I really ought to be able to understand this, and so it may be likely that someone reading this article in the future would get similarly confused.

Furthermore, the nudging term in the thermal budget is not mentioned at all in the text of section 4.2, and as with the momentum is also of similar character and opposite sign to the resolved tendency, particularly at the lowest pressures.

**L330-331: Is this conclusion necessarily true? ... (now L413)**

... Similarity between residual term error and total tendency error, which is proportional to the error of the corresponding variable itself as shown in the equation described above, suggests that the error of the corresponding variable itself stems from the residual term error ...

I can agree that this would be true, however, as before, I do not see similarity between residual term error and total tendency error in figure 4. And, you also seem to acknowledge that this similarity is not there, in line 450.

---

## Author Response (AR2)

Thank you so much for your insightful comments.

These comments are helpful to make the manuscript better and easier for readers.

Our responses to your comments are written below.

**As I understand it, the residual term for CNTL...**

... However, they are implicitly linked through their governing equations, with different assumption and approximation as indicated by referee #1. We think it is inappropriate to consider the two diagnostics to be completely independent.

...

Thank you for clarifying the agreement on the points that the tendencies and budgets are linked due to the governing equations, and that it is inappropriate to consider them independent. This did not come across in the original text and is useful to have clarified. The only further note I have on section 2.5 is for line 259, I might suggest changing "but is not necessarily true by definition" to "but they are not necessarily exactly equivalent". I assume based on lines 260-266 that this is the intended meaning, but as written I do not think the sentence means that. As written is seems something of a contradiction, as is the derivation of the budget not also the definition of the components?

Thank you for your proposal for an alternative description. As you indicated, the additional explanation of the description "but is not necessarily true by definition" follows in lines 260-266. I agree with you that "but they are not necessarily exactly equivalent" is more appropriate to explain the potential discrepancy between the budget diagnostics and the model tendencies. We have changed "but is not necessarily true by definition" to "but they are not necessarily exactly equivalent" for line 259.

A consequence of this is: taking the mean CNTL-GLN tendency from dynamics (4b, 6b, 8b, 10b), will this not be disproportionately dominated by the model dynamics bias in the first 6 hours being balanced by the nudging...

It may be the case that I was very unclear in my first review, and phrased this badly. I do not think it is adequately addressed as to why the CNTL-GLN nudging in figure 6(h) appears to have a similar character but with opposite sign to the dynamics in figure 6(f), and whether this is important to the conclusions.

The reason why the CNTL-GLN nudging in Fig. 6(h) appears to have a similar character but

with opposite sign to the dynamics in Fig. 6(f) will be explained below.

The similarity is noted in line 402, but in lines 413-414 you write that nudging increments are comparable to forecast errors. The forecast error in figure 1(c, e) has a similar character to the total tendency error in 4(d) and 6(e), however these do not have similar character to the resolved processes error (4e), the Coriolis error (5h) or the dynamics error (6f), which all instead are comparable to the nudging error (6h). Therefore it does not seem that nudging increments are comparable to forecast errors.

Figure 1(c,e) exhibits the mean error of the snapshot at Day 5 and 10 respectively, which are the medium-range timescale. The total tendency error averaged from Day 0 to Day 5 (Fig. 4(d) against GLN) is almost proportional to the error of the snapshot at Day 5 as indicated by Eq. (4). The structures and/or signs of the error could change within the first 5 days as illustrated by Fig. 2(d). Therefore, the error structure/sign in very-short-range timescale (e.g., 6 hours) does not necessarily last up to the medium-range timescale (e.g., 5 days).

Lines 413-414 mention that the nudging increments are comparable to the forecast errors "at least those growing over the 6-hour timescale", which is the very short-range timescale. The rationale behind this description is the nudging forcing formulation, the second term in the RHS of Eq. (1). The nudging forcing relaxes the model variables towards the analysis and reduces the error at the very end of each model timestep. The nudging forcing is proportional to the instantaneous error,  $X_M - X_A$ , at that point (similar to an analysis increment in data assimilation). Therefore, as you mentioned, we acknowledge that nudging increments are not necessarily comparable to the forecast errors themselves at the medium-range timescale. The result that the forecast error in Fig. 1(c,e) does not have similar character to the quasi-error in the resolved processes (Fig. 4(e)) or the unresolved processes (Fig. 4(f)) implies that the forecast error results from a balance/compensation between the resolved processes error and the unresolved processes error.

It is acknowledged in line 450 that the thermal budget and the momentum budgets are different. Here it is noted that there is similarity between the residual (8f) and the total (8d). This similarity is used to conclude deficiencies in physics parametrizations from the thermal budget. However by using the phrase "unlike the momentum budget", this highlights my confusion that you also conclude for the momentum budget that it is a deficiency in the physics parametrizations in lines 413-417, with similar phrasing, despite the opposite result. I interpret line 415-417 as saying that the model physics should be applying the same forcing that the nudging is, because you are using a nudged forecast as the truth.

A first look at figure 6 had me confused, as it seemed that the majority of the forcing of the nudging to move the GLN field towards the analysis was being directly un-done and opposed by the tendencies of the model dynamics. The majority of the nudging tendency being directly counteracted seemed to be an odd result, but as I have not personally used nudging in the same way as you are here in this study, it would be very useful if you could please provide your interpretation of what the cancellation of terms in figure 6(f) and 6(h) means, in terms of what the nudging is doing, how the model is responding, and how the differences between CNTL and GLN should be interpreted in light of this.

I feel like I really ought to be able to understand this, and so it may be likely that someone reading this article in the future would get similarly confused.

Furthermore, the nudging term in the thermal budget is not mentioned at all in the text of section 4.2, and as with the momentum is also of similar character and opposite sign to the resolved tendency, particularly at the lowest pressures.

According to Eq. (2) and (3), the residual terms are interpreted as an estimation of Friction/diabatic heating or parameterized forcing terms (i.e., model physics). Figure 4(c) looks like Fig. 6(c), which suggests that estimated offline unresolved tendency shown in Fig. 4(c) could approximate online physics parameterization forcing in CNTL. However, with respect to their difference between CNTL and GLN, Fig. 6(g) shows little difference, and Fig. 4(f) looks like Fig. 6(h), which exhibits nudging tendency in GLN with the reversed sign. We acknowledge this, and this can be accounted for by decomposing the error in physics parametrization into atmospheric fields and scheme. In a sense, a particular physics parametrization to calculate tendencies is considered as a multivariable function as follows:

$$\left(\frac{\overline{\partial u}}{\partial t}\right)_{\text{Phy,CNTL}} = F_{u,\text{Phy,Mdl}}(\phi_{1,\text{CNTL}}, \dots, \phi_{N,\text{CNTL}})$$

where  $F_{\text{Phy,Mdl}}$  denotes a given parameterization scheme employed by the model (in other words, one subroutine in Fortran code). As described in L296-L301 in Section 2.6, the scheme FPhv,Mdl used in CNTL and GLN is identical, while the input of the scheme such as background atmospheric fields is different between  $\phi_{n, \, \text{CNTL}}$  in CNTL and  $\phi_{n, \, \text{GLN}}$  in GLN. However, the scheme  $F_{Phv,Mdl}$  itself must have deficiencies against unknown truth of physics parameterization scheme  $F_{\text{Phy,Truth}}$ . An actual error in tendencies calculated by physics parameterizations we would like to know can be described as follows:

$$\left(\frac{\partial u}{\partial t}\right)_{\text{Phy,Error}} = F_{u,\text{Phy,Mdl}}(\phi_{1,\text{CNTL}}, \dots, \phi_{N,\text{CNTL}}) - F_{u,\text{Phy,Truth}}(\phi_{1,\text{Truth}}, \dots, \phi_{N,\text{Truth}})$$

$$\neq F_{u,\text{Phy,Mdl}}(\phi_{1,\text{CNTL}}, \dots, \phi_{N,\text{CNTL}}) - F_{u,\text{Phy,Mdl}}(\phi_{1,\text{GLN}}, \dots, \phi_{N,\text{GLN}})$$

The second line of this equation,  $F_{\text{Phy,Mdl}}(\phi_{1,\text{CNTL}},...,\phi_{N,\text{CNTL}}) - F_{\text{Phy,Mdl}}(\phi_{1,\text{GLN}},...,\phi_{N,\text{GLN}})$ ,

indicates what is shown in Fig. 6(g). Little difference in physics parameterization tendency between CNTL and GLN, shown in Fig. 6(g), implies that  $F_{\text{Phy,Mdl}}$  could be insensitive to differences in the inputs of the parameterization  $\phi_n$ . Furthermore, this implies that  $F_{\text{Phy,Mdl}}$  itself has deficiencies against unknown  $F_{\text{Phy,Truth}}$ . For this reason, Fig. 6(g) is not a good estimate of physics errors, and thus we diagnose the budget to estimate actual physics tendency using GLN experiment and estimate physics errors  $\left(\frac{\partial u}{\partial t}\right)_{\text{Phy,Error}}$  in Fig. 4(f). In other words, provided that the physics scheme is very poor (it is expected to be the case), the physics tendency calculated using the scheme should have a large error even if accurate

In terms of L450 versus L413-417, physics parameterization schemes  $F_{\text{Phy,Mdl}}$  have deficiencies against  $F_{\text{Phy,Truth}}$  and cause substantial errors in the short-range timescale (e.g., earlier than Day 1) for both zonal wind and temperature. Meanwhile, the similarity in temperature suggests that the error caused by the physics is significant even at Day 5 whereas the non-similarity in zonal wind suggests that the error at Day 5 results from a balance/compensation between the physics error and the dynamical response to the errors.

On the other hand, a dynamical core itself generally has smaller deficiencies than a physics parametrization scheme because the governing equations are basically well established. In addition, model dynamics could be more sensitive to the atmospheric fields than model physics as the governing equation itself suggests - for example, wind fields strongly affect the advective term or the Coriolis term.

What is exhibited in Fig. 6(e-h) is written as follows:

atmospheric fields are fed into the scheme.

$$\begin{split} \left(\frac{\partial u}{\partial t}\right)_{\text{Total,CNTL}} - \left(\frac{\partial u}{\partial t}\right)_{\text{Total,GLN}} \\ &= \left(F_{u,\text{Dyn,Mdl}}(\phi_{1,\text{CNTL}}, \dots, \phi_{N,\text{CNTL}}) - F_{u,\text{Dyn,Mdl}}(\phi_{1,\text{GLN}}, \dots, \phi_{N,\text{GLN}})\right) \\ &+ \left(F_{u,\text{Phy,Mdl}}(\phi_{1,\text{CNTL}}, \dots, \phi_{N,\text{CNTL}}) - F_{u,\text{Phy,Mdl}}(\phi_{1,\text{GLN}}, \dots, \phi_{N,\text{GLN}})\right) \\ &+ \left(-F_{u,\text{Ndg,Mdl}}(u_{\text{GLN}})\right) \end{split}$$

Figure 6(e,f,g,h) indicates the LHS, the first term, the second term, and the third term in the RHS, respectively. As mentioned above, the second term in the RHS is relatively small because of the physics scheme's insensitivity to the atmospheric fields. The LHS depends on forecast lead time until the error saturates. When the LHS and the second term in the RHS are small,

the first term in the RHS should have similar structures with the opposite sign to the nudging forcing to close the tendency budget.

Although further investigations are needed for justification, a scientific rationale of similar structures with the opposite sign between model dynamics response (Fig. 6(f)) and the nudging forcing (Fig. 6(h)) attempts to be explained here with an extremely simplified example. Since Fig. 5(h) indicates that the dynamics tendency difference (Fig. 6(f)) stems from Coriolis forcing difference between CNTL and GLN, the Coriolis response to the nudging forcing in GLN is addressed here. If positive zonal wind forcing were exerted by the nudging, negative zonal wind error in CNTL (imagine too strong parameterized drag, for example) would be removed in GLN. This change in zonal wind velocity leads to larger southward (northerly) Coriolis forcing in meridional momentum than CNTL, and then meridional wind velocity should decrease in GLN relative to CNTL. The decreased meridional wind is expected to cause weaker eastward (westerly) or stronger westward (easterly) Coriolis forcing. This response of Coriolis forcing to the positive nudging forcing is interpreted as a counteraction, which accounts for the opposite sign of the model dynamics against the nudging forcing.

Another simplified example for temperature may also be able to account for the dynamical response (Fig. 10(f)) to the nudging forcing (Fig. 10(h)). Since Fig. 9(d) indicates that the dynamics tendency difference (Fig. 10(f)) is dominated by the mean stationary flow component, mean flow response to the nudging forcing is focused on. If positive temperature forcing would be exerted by nudging (i.e., artificial heating), negative temperature error in CNTL should be removed in GLN. This change in temperature causes vertical motions relative to CNTL and then the artificial heating is expected to be compensated by diabatic cooling with ascents or mean flow advection. This is consistent with the opposite sign of the model dynamics against the nudging forcing in temperature.

This explanation may be too simplified, and it might be not necessarily the case in the model. Please keep in mind that there must be more complex interactions in GLN among model dynamics, model physics, and artificial nudging forcing.

To summarize what happens in GLN, nudging forcing reduces errors caused mainly by deficient model physics at very short-range timescale, and then dynamical core responds to constrained atmospheric fields by the artificial forcing without physical consistency to counteract nudging forcing whereas model physics doesn't respond to the forcing strongly. In the manuscript, how the errors are generated in CNTL against GLN is addressed for better understanding of error sources. In L413-418 for zonal wind and L450-455 for temperature,

we intend to explain our interpretations from CNTL's error perspective (i.e., a dynamical response in CNTL to the error) rather than what happens in GLN (i.e., a dynamical response in GLN to the nudging forcing which looks like the error with the reversed sign), while we think they indicate one thing seen from different perspectives (either 'CNTL minus GLN' shown in this paper or 'GLN minus CNTL').

With respect to thermal budgets and tendencies (Figs. 8-11), some of the physics parameterizations (especially longwave radiation scheme) could be more sensitive to the atmospheric fields than those for zonal momentum. In terms of the opposite impacts of the parameterization (Fig. 10(g)) relative to the total tendency (Fig. 10(e)) in the upper troposphere, a decomposition of the physics parameterization difference into the individual processes (i.e., convection, radiation) suggests that the longwave radiation scheme seems to generate more cooling with constrained higher temperature fields and less cooling with constrained lower temperature fields (see Fig. B1 below). We suppose that this could be accounted for by Stefan–Boltzmann law as first approximation, although what actually happens in the longwave radiation scheme is much more complex. This implies that the longwave radiation scheme consisting of physics parameterizations works to counteract a part of the temperature change.

Fig. B1 Latitude-height cross section of differences in the residual term of the zonal-mean thermal budget equation (the leftmost column) and zonal-mean temperature tendencies of individual processes in (Top) CNTL, (Middle) GLN, (Bottom) CNTL-GLN. The panel with black bold box shows longwave radiation tendencies. Note that the colour scale and the plotted uppermost altitude are different from those in Fig. 10 in the manuscript.

Hopefully the explanation above will clear up reviewer 1's questions.

If we added some explanations about what happens in GLN, descriptions from two perspectives, CNTL and GLN, coexist, and thereby they may bring some confusion. To avoid additional confusion, we would rather not include detailed explanations about what happens in GLN and how dynamical core and physics parameterizations response to the nudging forcing in the manuscript.

L417-420: Less usefulness of the tendency diagnostics to estimate correct unresolved physics tendency is described.

L453-461: The description on temperature error is improved, and the nudging forcing on temperature is mentioned.

**L330-331: Is this conclusion necessarily true? ... (now L413)**

... Similarity between residual term error and total tendency error, which is proportional to the error of the corresponding variable itself as shown in the equation described above, suggests that the error of the corresponding variable itself stems from the residual term error ... I can agree that this would be true, however, as before, I do not see similarity between residual term error and total tendency error in figure 4. And, you also seem to acknowledge that this similarity is not there, in line 450.

In general, even if some physics parameterization have deficiencies ( $F_{\text{Phy,Mdl}}$  versus  $F_{\text{Phy,Truth}}$ ) and the residual term causes substantial errors, the errors caused by model dynamics (including a correct dynamical response to incorrect unresolved forcing) may hide the errors caused by model physics parameterizations in longer forecast lead time. It is quite important to disentangle errors and decompose their sources into processes.

As you mentioned, we acknowledge that we do not see similarity between the residual term error shown in Fig. 4(f) and the total tendency error shown in Fig. 4(d). It is the forecast errors "at least those growing over the 6-hour timescale" that the nudging increments are comparable to in this study, as suggested by Eq. (1) or the nudging formulation. The error structure/sign in very short-range timescale (e.g., 6 hours) does not necessarily last up to medium-range timescale (e.g., 5 days). We have intended to demonstrate that the error structure is similar in temperature field between these two different timescales but not in zonal wind field. As you seem to acknowledge, the residual term error is similar to the total tendency error relatively in temperature (Fig. 8(f) versus Fig. 8(d)), but not similar in zonal wind velocity (Fig. 4(f) versus Fig. 4(d)). This is the reason why we conclude that temperature error in the medium-range timescale (Fig. 1(d)) is caused mainly by deficient physics parameterizations and zonal wind error in the timescale (Fig. 1(c)) results from the error compensation between deficient physics parameterizations and incorrect dynamical responses.